# Learning Adaptive Multiresolution Transforms via Meta-Framelet-based Graph Convolutional Network

**Tianze Luo**[*1]  **Zhanfeng Mo**[*1]  **Sinno Jialin Pan**[1,2]
[1] Nanyang Technological University, Singapore;   [2] The Chinese University of Hong Kong
{tianze001, zhanfeng001}@ntu.edu.sg, sinnopan@cuhk.edu.hk

## Abstract

Graph Neural Networks are popular tools in graph representation learning that capture the graph structural properties. However, most GNNs employ single-resolution graph feature extraction, thereby failing to capture micro-level local patterns (high resolution) and macro-level graph cluster and community patterns (low resolution) simultaneously. Many multiresolution methods have been developed to capture graph patterns at multiple scales, but most of them depend on predefined and hand-crafted multiresolution transforms that remain fixed throughout the training process once formulated. Due to variations in graph instances and distributions, fixed handcrafted transforms can not effectively tailor multiresolution representations to each graph instance. To acquire multiresolution representation suited to different graph instances and distributions, we introduce the **M**ultiresolution **M**eta-**F**ramelet-based **G**raph **C**onvolutional **N**etwork (**MM-FGCN**), facilitating comprehensive and adaptive multiresolution analysis across diverse graphs. Extensive experiments demonstrate that our MM-FGCN achieves SOTA performance on various graph learning tasks. The code is available on GitHub[1].

## 1 Introduction

The ubiquity of graph-structured data (Zhou et al., 2020; Wu et al., 2020; Sanchez-Gonzalez et al., 2018; Fout et al., 2017; Hamaguchi et al., 2017) in today's interconnected society has sparked immense interest in the machine learning community for processing and analysis of such data, which leverages mathematical representations like graphs to capture interdependencies between data entities. Graph neural network (GNN) has found widespread adoption due to ease of implementation and quality of prediction. Recent research (Balcilar et al., 2021; Geng et al., 2023) underscores that most GNN models, including GCN (Kipf & Welling, 2017), GAT (Thekumparampil et al., 2018), and GraphSage (Hamilton et al., 2017b), fundamentally operate as low-pass filters in the context of graph signal processing (Chang et al., 2021). They generate smooth node embeddings using low-resolution features, where neighboring graph nodes share similar graph features, and a local feature aggregation leads to informative representations.

However, capturing solely low-resolution information is insufficient for achieving a comprehensive graph representation. Low-resolution information represents graph signals that vary smoothly over the graph and are associated with low-frequency graph signals, whereas high-resolution information encompasses local disruption and detailed patterns that are associated with high-frequency graph signals. Thus, it is also crucial to capture the fine-grained graph details at high-resolution levels. For instance, GNNs may fail on disassortative graphs (Liu et al., 2022a; Pei et al., 2020; Suresh et al., 2021), where locally connected nodes often exhibit different features and labels. This heterogeneity emphasizes the necessity of using the high-pass graph filters to capture the disruptive local patterns (Liu et al., 2022a; Pei et al., 2020). In another example, for social network data, high- and low-frequency components represent micro and macro-level dynamics respectively. While the micro-level highlights individual interactions, revealing personal influences, the macro-level captures

---

[*]Both authors contributed equally to this research.
[1]https://github.com/ltz0120/graph-multiresolution-meta-framelet

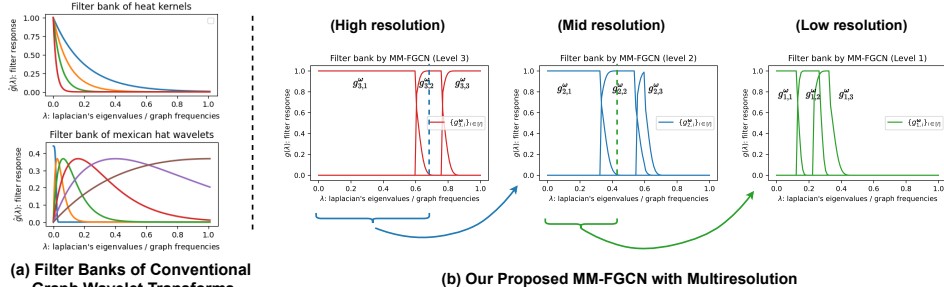

Figure 1: Comparison of the filter banks of the conventional graph wavelet transforms with our proposed MM-FGCN with learnable multiresolution filter banks. We plot three levels of resolutions and each resolution level contains one low-pass filter and two high-pass filters.

communities, clusters, and motifs, shedding light on broader social relations and group behaviors. Therefore, a GNN that relies solely on features from one or a few resolution levels fails to capture a comprehensive graph representation, necessitating the use of multiresolution graph analysis.

Recent advancements in multiresolution graph representation learning fall into two main categories, i.e. the 1) graph-structure-based approach (González & Ortega, 2019; Bacciu et al., 2023; Geng et al., 2023; Xu et al., 2019c), which usually adopts down-sampling methods to partition the graph into multiple resolutions, or adopt specially designed GNN such as Graph-U-Net (Gao & Ji, 2019) to capture the graph features at different resolutions. 2) Graph-spectral-based approach, where some of the methods under this category adopt low-pass and high-pass spectral filters (Zhu et al., 2021; Chien et al., 2020). Other methods adopt wavelet transforms (Zheng et al., 2021b;a) to project graph signals to graph signal subspaces of different resolution levels. The wavelet frame transform provides an efficient way to obtain representations based on features of various scales.

Most current multiresolution methods rely on either heuristic, inflexible spatial down- and up-sampling strategies, or fixed, manually crafted spectral filters. For instance, the MR-GNN model (Xu et al., 2019c) employs multi-hop convolution layers with receptive fields of a fixed size. UFGConv (Zheng et al., 2021a) and WFTG (Dong, 2017b) leverage deliberately designed graph framelet transform to discern graph signals across various resolutions. Furthermore, PyGNN (Geng et al., 2023) utilizes a manually devised downsampling technique to categorize graph signals into different frequency levels. However, the reliance of these methods on fixed multiresolution analysis strategies imposes significant limitations on obtaining high-performing representations. In practice, various graph instances and distributions may manifest distinct scales and resolution factors, with their discriminative information residing at different resolution levels. Additionally, designing an appropriate multiresolution transform demands a deep understanding of the dataset-specific inductive bias, making it hard to generalize to other domains. Thus, employing fixed multiresolution analysis strategies fails to customize an appropriate multiresolution transform for individual graph instances.

To address this limitation, it is crucial to learn an adaptive multiresolution representation that can be automatically tailored to diverse graph instances and distributions. This motivates us to establish a meta-learner to generate the customized feature transform and multiresolution analysis strategy for each individual graph instance. In this paper, we introduce the **M**ultiresolution **M**eta-**F**ramelet-based **G**raph **C**onvolution **N**etwork (**MM-FGCN**), a spectral-based method designed to learn adaptive multiresolution representations for different graph instances. For each input graph instance, the MM-FGCN first generates the meta-framelet generator, which consists of a set of customized band-pass filters in the frequency domain. The meta-framlet generator in turn induces a set of framelet-based multiresolution bases. Then, the input graph feature is decomposed into multiresolution components through projections onto each multiresolution basis. Finally, these multiresolution components are manipulated and passed to successive layers for downstream feature processing. As visualized in Figure 1, our MM-FGCN creates an adaptive multiresolution transform for each graph instance by learning a customized stratified multiresolution frequency partition in the frequency domain. In contrast, traditional filter-based and wavelet-based methods are confined to employing a fixed multiresolution analysis strategy across the entire graph dataset.

**Contributions.** In this paper, we propose a novel MM-FGCN for adaptive multiresolution representation learning. The contribution of this paper is three-fold.

- We introduce Multiresolution Meta-Framelet System (MMFS) (Section 4.1), a set of learnable multiresolution bases that can be simply constructed based on a set of meta-band-pass filters (Section 4.2).

- We show that MMFS induces a series of progressive resolution graph signal spaces that inherently possess denseness, tightness, and dilation and translation properties (Section 4.1). Thus, the multiresolution decomposition and reconstruction for any graph signal can be achieved by projections onto each basis in MMFS.

- Based on the MMFS-based multiresolution transform, we propose the Multiresolution Meta-Framelet-based Graph Convolutional Network (MM-FGCN) (Section 4.3) for adaptive multiresolution graph signal processing. Extensive experiments show that our model achieves state-of-the-art performance compared to other baseline methods.

## 2 RELATED WORK

**Multi-resolution Graph Representation Learning.** Graph representation learning with multi-resolution techniques aims to provide multi-scale views of the graph data to better understand the local/detailed and global/overall information. Conventional methods adopt the techniques from computer visions, constructing different views based on the graph structure. González & Ortega (2019) adopt the downsampling method to retrieve the graph at different resolutions and perform graph matching. MR-GNN (Xu et al., 2019c) adopt several weighted graph convolution layers to learn graph representations at different granularities. Geng et al. (2023) propose the Pyramid Graph Neural Network which converts graphs with multi-scale downsampling. Another stream of graph multiresolution analysis focuses on designing graph spectral filters Dong (2017b); Mallat (1989) to decompose graph signals into multiple resolutions. Zheng et al. (2021a) introduce a tight graph framelet system with multiresolution, to deliberately process graph signals at different scales. FAGCN (Bo et al., 2021) enhances graph convolution with frequency adaptation that integrates different frequencies via an attention mechanism, and GPR-GNN (Chien et al., 2020) iteratively combines multiple frequencies with generalized pagerank. However, the capability of the aforementioned model is limited by the fixed spectral filter.

## 3 PRELIMINARY

We focus on undirected graphs represented as $G = (\mathbf{X}, \mathbf{A})$ with $n$ nodes. Here, $\mathbf{X} \in \mathbb{R}^{n \times d}$ represents node features, and $\mathbf{A} \in \mathbb{R}^{n \times n}$ is the adjacency matrix, where $\mathbf{A}[i, j] > 0$ if an edge exists between node $i$ and $j$, and $\mathbf{A}[i, j] = 0$ otherwise. The Laplacian matrix of the graph is defined as $\mathbf{L} = \mathbf{D} - \mathbf{A}$, where $\mathbf{D} = \mathrm{diag}(\mathbf{A}\mathbf{1}_n)$ is the diagonal degree matrix with $\mathbf{D}[i, i] = \sum_{i=1}^{n} \mathbf{A}[i, j]$, and $\mathbf{1}_n$ is an all-one vector of size $n$. Without specification, $\langle \cdot, \cdot \rangle$ denotes the inner product, $[n]$ denotes $\{1, ..., n\}$.

**Graph Representation Learning.** For any graph data $G$ sampled from the graph domain $\mathcal{G}$, graph representation learning aims to learn a graph representation $f_{\boldsymbol{\theta}}(\cdot) : \mathcal{G} \mapsto \mathbb{R}^{n \times h}$, with which we can embed each node of $G$ into a $h$-dimensional compact vector, facilitating downstream tasks such as node classification and graph classification. A desirable graph representation should be able to capture the essential graph structural properties.

**Spectral Graph Signal Processing.** A graph signal $x(\cdot)$ generally refers to a $G \mapsto \mathbb{R}$ mapping. As $x(\cdot)$ assigns a value to each of the $n$ nodes, it is represented by a vector $\mathbf{x} \in \mathbb{R}^n$, where $\mathbf{x}[i]$ corresponds to the graph signal value assigned to the $i$-th node of $G$. In spectral graph signal processing (Kipf & Welling, 2017; Shuman et al., 2013), the graph Laplacian $\mathbf{L}$ plays a crucial role in graph modeling and analysis, and it is tightly related to graph structural properties, including clusterability (Chiplunkar et al., 2018), connectivity (Fiedler, 1989), node distance, etc. In fact, $\mathbf{L}$ serves as a graph shift operator which enables us to transform a graph signal into the frequency domain and manipulate its frequency components. Suppose the eigendecomposition of the graph Laplacian is $\mathbf{L} = \mathbf{U}\boldsymbol{\Lambda}\mathbf{U}^{\top}$, the *graph spectrum* refers to the diagonal eigenvalue matrix $\boldsymbol{\Lambda} = \mathrm{diag}(\lambda_1, ..., \lambda_n)$, and the *spectral bases* is the collection of eigenvectors $\mathbf{U} = (\mathbf{u}_1, ..., \mathbf{u}_n)$. Thus, a graph signal $\mathbf{x}$ can be transformed into the frequency domain via graph Fourier transform $\widehat{\mathbf{x}} = (\langle \mathbf{u}_1, \mathbf{x} \rangle, ..., \langle \mathbf{u}_n, \mathbf{x} \rangle)^{\top} = \mathbf{U}^{\top}\mathbf{x}$, and it can be reconstructed from its frequency components $\hat{\mathbf{x}}$ via the inverse graph Fourier transform $\mathbf{x} = \sum_i \langle \mathbf{u}_i, \mathbf{x} \rangle \mathbf{u}_i = \mathbf{U}\widehat{\mathbf{x}}$. Furthermore, one can apply a smooth filter $g_{\boldsymbol{\theta}}$ to manipulate

frequency components of $\mathbf{x}$ by the *spectral convolution* (Kipf & Welling, 2017)

$$g_{\boldsymbol{\theta}}(\mathbf{L}) * \mathbf{x} \triangleq \mathbf{U}g_{\boldsymbol{\theta}}(\boldsymbol{\Lambda})\mathbf{U}^{\top}\mathbf{x}.$$

In machine learning practice, applying spectral convolution to the graph feature $\mathbf{X}$ (which can be viewed as a $d$-dimensional graph signal) provides us with informative graph representation. Different implementations of filter $g_{\boldsymbol{\theta}}$ lead to desirable graph representations for different purposes, such as classification, denoising, smoothing, and abnormally detection (Xu et al., 2019a; Gasteiger et al., 2019; Li et al., 2021a; Tang et al., 2022).

**Spectral Graph Multiresolution Analysis.** Classic multiresolution analysis (Mallat, 1989; Cohen et al., 1993) aims to decompose a signal into multiple components of varying resolutions, which can then be processed individually to provide a comprehensive representation of the signal. Let $L^2(\mathbb{R})$ be the measurable, square-integrable one-dimensional functions, where the inner product of $x, z \in L^2(\mathbb{R})$ is $\langle x, z \rangle = \int x(t)z(t)\mathrm{d}t$. Given a resolution factor $\gamma > 1$, the multiresolution decomposition for signals in $L^2(\mathbb{R})$ is determined by a series of progressive resolution function spaces $\{V_r\}_r$, where each $V_r$ is a subspace of $L^2(\mathbb{R})$, and $V_r \subset V_{r'}$ if $r < r'$. The $\{V_r\}_r$ is expected to satisfy the *denseness*, *dilation property*, and *translation property* (Mallat, 2006), ensuring that $V_r$ collects the $\gamma^r$-resolution signals, and the multiresolution decomposition of any given signal $x$ can be achieved by projecting it into each subspace $V_r$.

- *Denseness*: $\{V_r\}_r$ contains sufficient information to represent and reconstruct any signal, that is, the union of $\{V_r\}_r$ is dense in $L^2(\mathbb{R})$, and the intersection of $\{V_r\}_r$ is $\{0\}$.

- *Dilation property*: signals in $V_r$ can be derived from signals in $V_{r+1}$ by scaling them using a resolution factor of $\gamma$, that is, $\psi(t) \in V_r \iff D_\gamma\psi(t) = \psi(\gamma t) \in V_{r+1}$, where $D_\gamma$ is the dilation operator.

- *Translation property*: when a signal $x$ is translated for $s$ in the spatial domain, its $\gamma^r$-resolution component translates for the same amount in the frequency domain, that is, $P_r(T_s x) = T_s P_r(x)$, where $P_r : L^2(\mathbb{R}) \mapsto V_r$ is the projection to $V_r$, and $T_s x(\cdot) = x(s - \cdot)$ is the translation operator.

The goal of multiresolution analysis is to determine a set of bases $\{\psi_{ri}\}_i$ that spans the desirable $V_r$, satisfying the denseness, dilation, and translation properties. Moreover, the $\gamma^r$-resolution component of a signal $x$ should be derivable from its projection onto each basis, i.e. $P_r(x) = \sum_i \langle \psi_{ri}, x \rangle \psi_{ri}$. Thus, the multiresolution decomposition of $x$ can be achieved by $x = \sum_r P_r(x) = \sum_{r,i} \langle \psi_{ri}, x \rangle \psi_{ri}$. For instance, a proper choice of $V_r$ is the collection of piecewise constant functions over $[-\gamma^r, \gamma^r]$, and $\psi_{ri}$ can be set as the associated Haar-like wavelets (Dong, 2017b).

For multiresolution graph analysis, one needs to extend the dilation and translation properties to the graph signal domain (where each graph signal is represented by a vector in $\mathbb{R}^n$) and determine the multiresolution graph bases $\{\boldsymbol{\varphi}_{ri}\}_{r,i} \subset \mathbb{R}^n$. To this end, one needs to define the spatial dilation and translation operators for graph signals by generalizing the scalar multiplication and node subtraction to the graph domain. According to the harmonic analysis theory (Stein, 1993; Gavish et al., 2010), the graph dilation and translation operators can be defined based on the graph Fourier transform. Consider a graph signal $\boldsymbol{\varphi} \in \mathbb{R}^n$ generated by a one-dimensional filter $g$, i.e. $\boldsymbol{\varphi} = \sum_k g(\lambda_k) \mathbf{u}_k$, the $\gamma$-dilation and $v$-translation of $\boldsymbol{\varphi}$ are defined as

$$D_\gamma \boldsymbol{\varphi} = \sum_k g(\gamma\lambda_k) \mathbf{u}_k, \ \forall \gamma > 0, \ \text{and} \ T_v \boldsymbol{\varphi} = \sum_k g(\lambda_k)\mathbf{u}_k[v] \mathbf{u}_k, \ \forall v \in G,$$

respectively. Therefore, finding the desirable multiresolution bases is equivalent to identifying a set of filters $\{g_{ri}\}_{r,i}$ such that the bases $\{\boldsymbol{\varphi}_{riv}\}_{r,i,v}$ generated by $\boldsymbol{\varphi}_{riv} = \sum_k g_{ri}(\lambda_k)\mathbf{u}_k[v] \mathbf{u}_k$ satisfies the aforementioned conditions.

Finally, a desirable set $\{\boldsymbol{\varphi}_{riv}\}_{r,i,v}$ must exhibit *tightness*. The set of bases is called *tight* if and only if $\|\mathbf{x}\|^2 = \sum_{r,i,v} |\langle \boldsymbol{\varphi}_{riv}, \mathbf{x} \rangle|^2$ holds for arbitrary $\mathbf{x}$. Intuitively, tightness ensures that the projection operator onto these bases preserves the overall energy (norm) of the original graph signal. It's worth noting that this property, while essential, is less restrictive than orthogonality. As guaranteed by the polarization identity, it enables multiresolution decomposition via $\mathbf{x} = \sum_{r,i,v} \langle \boldsymbol{\varphi}_{riv}, \mathbf{x} \rangle \boldsymbol{\varphi}_{riv}$.

This decomposition can be equivalently expressed as $\mathbf{x} = \sum_{r,i,v} \boldsymbol{\varphi}_{riv}\boldsymbol{\varphi}_{riv}^{\top} \mathbf{x} = \boldsymbol{\Phi}\boldsymbol{\Phi}^{\top}\mathbf{x}$, where $\boldsymbol{\Phi}$ is an $n$-by-$N$ *multiresolution transform matrix*, with each column representing a basis $\boldsymbol{\varphi}_{riv}$, and $N$ is the total number of bases. As the multiresolution transform matrix is defined by concatenating the multiresolution bases, we will use these two terms interchangeably throughout the rest of the paper.

# 4 METHODOLOGY

We propose the Multiresolution Meta-Framelet-based Graph Convolution Network (MM-FGCN), designed for adaptive multiresolution representation learning for varying graph instances. In Section 4.1 and Section 4.2, we construct a set of learnable multiresolution bases, termed Multiresolution Meta-Framelet System (MMFS). Our MMFS inherently possesses tightness and spans progressive multiresolution graph signal subspaces that satisfy denseness, dilation, and translation properties. For each graph, MM-FGCN first calculates the adaptive MMFS and the associated multiresolution transform matrix. This matrix enables us to decompose and manipulate the multiresolution components of the graph feature, yielding comprehensive graph representations (Section 4.3).

## 4.1 MULTIRESOLUTION META-FRAMELET SYSTEM

As mentioned in Section 1, learning the adaptive multiresolution bases is essential for obtaining a comprehensive graph representation. Suppose $N$ is the total number of multiresolution bases, a straightforward approach is to learn the multiresolution transform matrix via a neural network $\mathcal{M}_{\boldsymbol{\xi}} : \mathcal{G} \mapsto \mathbb{R}^{n \times N}$ parameterized by $\boldsymbol{\xi}$, such that $\boldsymbol{\Phi} = \mathcal{M}_{\boldsymbol{\xi}}(\mathbf{X}, \mathbf{A})$. However, without additional constraints, this directly learned $\boldsymbol{\Phi}$ may fail to meet the tightness property $\boldsymbol{\Phi}\boldsymbol{\Phi}^{\top} = \mathbf{I}$, making the multiresolution decomposition infeasible. Even if we impose constraints on $\boldsymbol{\Phi}$ to ensure tightness, denseness, translation, and dilation properties, the constrained optimization process becomes challenging to solve due to numerical instability. Additionally, learning a dense $n \times N$ matrix requires an excessive amount of parameters, leading to a significant computational overhead.

To address these limitations, we construct a set of learnable multiresolution bases with much fewer parameters, called the Multiresolution Meta-Framelet System (MMFS). MMFS consists of a set of learnable graph framelets, each generated by a spectral *meta-filter*. Individually, these meta-filters are distinguished by their trainable bandwidth parameters and specific resolution levels, all while sharing a common trainable resolution factor. The following arguments show that our MMFS is born to be tight, and it spans progressive multiresolution spaces that possess denseness, dilation, and translation properties. Hence, multiresolution decomposition can be achieved by using the MMFS-based multiresolution transform.

**Definition 1** (Multiresolution Meta-Framelet System). *Given the number of resolution levels $R > 0$, for the each resolution level $r \in [R]$, we define $I$ spectral meta-filters $\left\{g_{r,1}^{\boldsymbol{\omega}}, ..., g_{r,I}^{\boldsymbol{\omega}}\right\}$. These meta-filters are mappings from the interval $[0,1]$ to itself, and they are parameterized by a vector $\boldsymbol{\omega} \in \boldsymbol{\Omega}$. The collection of the $R \times I$ meta-filters is called the meta-framelet generator. We define the meta-framelet learner as $\mathcal{M}_{\boldsymbol{\xi}}(\cdot) : \mathcal{G} \mapsto \boldsymbol{\Omega} \times \mathbb{R}^{+}$, a neural network that maps any graph instance $G = (\mathbf{X}, \mathbf{A})$ to a specific meta-framelet generator $\boldsymbol{\omega}$ and a resolution factor $\gamma$. The Multiresolution Meta-Framelet System (MMFS) is defined as a set of graph signals $\left\{\boldsymbol{\varphi}_{riv}\right\}$, where*

$$\boldsymbol{\varphi}_{riv} = \sum_{k=1}^{n} g_{r,i}^{\boldsymbol{\omega}}\left(\gamma^{-J+r} \cdot \lambda_k\right) \mathbf{u}_k[v] \, \mathbf{u}_k, \tag{1}$$

*where $(\boldsymbol{\omega}, \gamma) = \mathcal{M}_{\boldsymbol{\xi}}(\mathbf{X}, \mathbf{A})$, $\lambda_k$ and $\mathbf{u}_k$ is the k-th eigenvalue and eigenvector of the graph Laplacian $\mathbf{L}$, and $J$ is the smallest value such that $\gamma^{-J+R}\lambda_{\max}(\mathbf{L}) \leqslant 1$. The MMFS-based multiresolution transform matrix is defined as the concatenation of $\left\{\boldsymbol{\varphi}_{riv}\right\}$, that is*

$$\boldsymbol{\Phi}_{\mathrm{MM}} \triangleq \left(\mathbf{U} \, g_{1,1}^{\boldsymbol{\omega}}(\gamma^{-J+1}\boldsymbol{\Lambda}) \, \mathbf{U}^{\top}, \cdots, \mathbf{U} \, g_{R,I}^{\boldsymbol{\omega}}(\gamma^{-J+R}\boldsymbol{\Lambda}) \, \mathbf{U}^{\top}\right). \tag{2}$$

Definition 1 illustrates the construction of MMFS based on the meta-framelet generator. Here, $\boldsymbol{\varphi}_{riv}$ represents the basis comprising a $r$-resolution dilation and translation w.r.t the $v$-th node. At the $r$-resolution level, the meta-filter $g_{r,i}^{\boldsymbol{\omega}}$ filtrates the information localized around the $v$-th node. Notably, equation 2 enables the efficient computation of $\boldsymbol{\Phi}_{\mathrm{MM}}$. This can be achieved by circumventing the need for eigen-decomposition of $\mathbf{L}$ through the application of Chebyshev approximation (Defferrard et al., 2016b) to $g_{r,i}^{\boldsymbol{\omega}}(\gamma^{-J+r}\mathbf{L})$. The subsequent proposition offers a construction for the meta-framelet generator, ensuring that the MMFS meets the criteria of tightness, denseness, translation, and dilation. The proof is available in Appendix D.

**Proposition 2** (MMFS-based Multiresolution Decomposition). *Following the notations in Definition 1, suppose the meta-framelet generator satisfies*

- $g^{\boldsymbol{\omega}}_{1,1}(\lambda)^2 + \cdots + g^{\boldsymbol{\omega}}_{1,I}(\lambda)^2 = 1, \forall \lambda \in [0,1]$.

- $g^{\boldsymbol{\omega}}_{r,i}(\gamma^{-J+r}\lambda) = g^{\boldsymbol{\omega}}_{1,i}(\gamma^{-J+r}\lambda)\, g^{\boldsymbol{\omega}}_{1,1}(\gamma^{-J+r-1}\lambda)\, \cdots\, g^{\boldsymbol{\omega}}_{1,1}(\gamma^{-J+1}\lambda), \forall r > 1, i \in [I]$,

*then following the construction in Definition 1, the MMFS induced by* $\{\boldsymbol{\varphi}_{riv}\}$ *forms a tight bases system. Here, the indices* $(r,i,v)$ *are iterated over* $v \in [n]$, *with* $(r,i)$ *drawn from the set* $([R] \times [I]) \backslash (r,1) : 1 \leqslant r < R$. *For any graph signal* $\mathbf{x} \in \mathbb{R}^n$, *the multiresolution transform matrix is* $\boldsymbol{\Phi}_{\mathrm{MM}} \in \mathbb{R}^{n \times (R(I-1)n)}$, *the multiresolution decomposition is achieved by*

$$\mathbf{x} = \sum_{r,i,v} \langle \boldsymbol{\varphi}_{riv}, \mathbf{x} \rangle\, \boldsymbol{\varphi}_{riv} = \boldsymbol{\Phi}_{\mathrm{MM}} \boldsymbol{\Phi}_{\mathrm{MM}}^{\top}\, \mathbf{x},$$

*where* $\mathbf{x} \mapsto \boldsymbol{\Phi}_{\mathrm{MM}}^{\top}\mathbf{x}$ *is the multiresolution transform. Moreover, let* $V_r = \mathrm{span}(\{\boldsymbol{\varphi}_{riv}\}_{i,v})$, *the resulting subspaces* $\{V_r\}_r$ *turn out to be a series of progressive resolution space that possess denseness, dilation, and translation properties.*

Proposition 2 shows that, once the meta-filters of the 1-resolution level are determined, a desirable MMFS can be constructed in a stratified and iterative manner. As visualized in Figure 1, the $r$-resolution level meta-filters $\{g^{\boldsymbol{\omega}}_{r,1}, ..., g^{\boldsymbol{\omega}}_{r,I}\}$ induce a unitary partition within the support of $g^{\boldsymbol{\omega}}_{r+1,1}$, which is the low-pass filter of the $(r+1)$-resolution level.

## 4.2 META-FRAMELET GENERATOR

To implement MMFS-based transform, the remaining step is to design the formulation of the meta-framelet generator $\{g^{\boldsymbol{\omega}}_{1,1}, ..., g^{\boldsymbol{\omega}}_{1,I}\}$ such that $\sum_i {g^{\boldsymbol{\omega}}_{1,i}}^2 \equiv 1$. This inspires us to simply set $\{g^{\boldsymbol{\omega}}_{1,1}, ..., g^{\boldsymbol{\omega}}_{1,I}\}$ as $I$ band-pass filters to partition the $[0,1]$ interval into $I$ regions. In this paper, we instantiate each $g^{\boldsymbol{\omega}}_{1,i}$ as a meta-band-pass filter based on polynomial splines in Han et al. (2016), i.e.

$$g^{\boldsymbol{\omega}}_{1,i}(\lambda) \triangleq \begin{cases} 0, & \lambda \in [0, c_{i-1} - \varepsilon_{i-1}] \cup [c_i + \varepsilon_i, 1], \\ \sin(\frac{\pi(\lambda - c_{i-1} + \varepsilon_{i-1})}{4\varepsilon_{i-1}}), & \lambda \in (c_{i-1} - \varepsilon_{i-1}, c_{i-1} + \varepsilon_{i-1}), \\ 1, & \lambda \in [c_{i-1} + \varepsilon_{i-1}, c_i - \varepsilon_i], \\ \cos(\frac{\pi(\lambda - c_i + \varepsilon_i)}{4\varepsilon_i}), & \lambda \in (c_i - \varepsilon_i, c_i + \varepsilon_i), \end{cases} \tag{3}$$

where $\{c_1, \varepsilon_1, ..., c_{I-1}, \varepsilon_{I-1}\}$ are parameters encoded in $\boldsymbol{\omega}$. Specifically, for any $\boldsymbol{\omega} \in \boldsymbol{\Omega} \subset \mathbb{R}^{2(I-1)}$, we define

$$c_i \triangleq \frac{1}{\|\boldsymbol{\omega}\|^2} \sum_{j \leqslant i} \boldsymbol{\omega}[j]^2,\ \varepsilon_i \triangleq \alpha \min\{c_i - c_{i-1}, c_{i+1} - c_i\}, \tag{4}$$

where $\alpha \in (0, 1/2)$ is a predefined hyperparameter and it holds that $0 = c_0 \leqslant c_1 \leqslant \cdots \leqslant c_{I-1} \leqslant c_I = 1$. Notably, the parameterization of the meta-framelet generator uses only $2(I-1)$ parameters, significantly reducing the budget compared to the dense $n$-by-$(R \times (I-1) \times n)$ multiresolution transform matrices. Intuitively, the meta-filters adaptively decompose graph features into spectral channels and process frequency components at various resolution levels, leading to a flexible and comprehensive graph representation.

## 4.3 MULTIRESOLUTION META-FRAMELET-BASED GRAPH CONVOLUTION NETWORK

Leveraging the efficient construction and computation of MMFS-based multiresolution transform matrix $\boldsymbol{\Phi}_{\mathrm{MM}}$ in Proposition 2, we can now establish the Multiresolution Meta-Framelet-based Graph Convolution (MM-FGConv) and its associated graph pooling counterpart (MM-FGPool). These operators serve as meta-analogs to the conventional graph convolution and graph pooling methods (Defferrard et al., 2016b). The MM-FGPool operator is simply defined as $\mathrm{MMFGPool}_{\boldsymbol{\xi}}(\mathbf{H}; \mathbf{X}, \mathbf{A}) \triangleq \mathbf{1}^{\top}\boldsymbol{\Phi}_{\mathrm{MM}}^{\top}\mathbf{H}$, where the meta-framelet coefficients are aggregated and concatenated as the output of the readout of the final classifier. The computation of MM-FGConv is illustrated in Figure 2 and its details are presented in Algorithm 1.

An $L$-layer MMFS-based Graph Convolutional Network (MM-FGCN) is defined by

$$\mathrm{MMFGCN}_{\boldsymbol{\theta}, \mathbf{w}; \boldsymbol{\xi}}(\mathbf{A}, \mathbf{X}) \triangleq h \circ \mathrm{MMFGPool}_{\boldsymbol{\xi}}(\mathbf{H}_L \mathbf{W}_L; \mathbf{A}, \mathbf{X}), \tag{5}$$

$$\mathbf{H}_l \triangleq \sigma\left(\mathrm{MMFGConv}_{\boldsymbol{\Theta}_{l-1}, \boldsymbol{\xi}}(\mathbf{H}_{l-1}; \mathbf{A}, \mathbf{X})\mathbf{W}_{l-1}\right), \forall l \in [L], \tag{6}$$

$$\boldsymbol{\theta} \triangleq \mathrm{vec}\left(\{\boldsymbol{\Theta}_l\}_{l \in [L]}\right),\ \mathbf{w} \triangleq \mathrm{vec}\left(\{\mathbf{W}_l\}_{l \in [L]}\right), \tag{7}$$

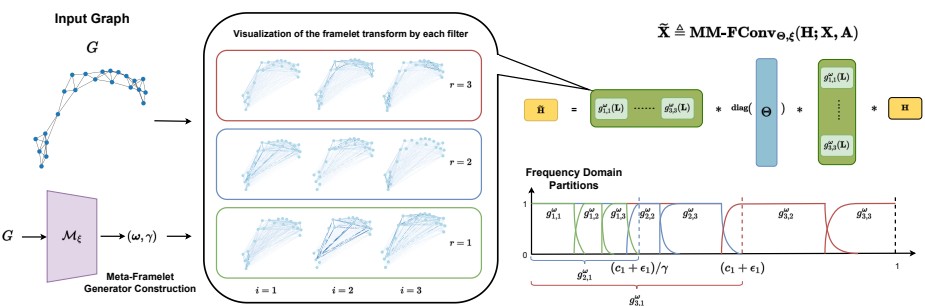

Figure 2: the computation of MM-FGConv operator with a meta-framelet learner $\mathcal{M}_{\boldsymbol{\xi}}$ and learnable filter $\boldsymbol{\Theta}$.

**Algorithm 1** MM-FGConv

1: **Input:** graph data $G = (\mathbf{X}, \mathbf{A})$, graph Laplacian $\mathbf{L}$, the meta-framelet generators $\{g_{r,i}^{\boldsymbol{\omega}}\}$, a meta-neural network $\mathcal{M}_{\boldsymbol{\xi}}$, a learnable diagonal matrix $\boldsymbol{\Theta}$, the $T$-order Chebyshev polynomial approximation $\text{Chebyshev}_T(\cdot)$.
2: **Output:** $\widetilde{\mathbf{X}}$, the output processed graph signal.
3: $J \leftarrow \lceil \log_\gamma(\lambda_{\max}(\mathbf{L})) + R \rceil$
4: $(\boldsymbol{\omega}, \gamma) \leftarrow \mathcal{M}_{\boldsymbol{\xi}}(\mathbf{X}, \mathbf{A})$
5: $\widetilde{g}_{r,i}^{\boldsymbol{\omega}} \leftarrow \text{Chebyshev}_T(g_{r,i}^{\boldsymbol{\omega}})$, for $r \in [R], i \in [I]$
6: $\boldsymbol{\Phi}_{\text{MM}} \leftarrow \left( \widetilde{g}_{1,1}^{\boldsymbol{\omega}}(\gamma^{-J+1}\mathbf{L}), ..., \widetilde{g}_{R,I}^{\boldsymbol{\omega}}(\gamma^{-J+R}\mathbf{L}) \right)$
7: $\widetilde{\mathbf{X}} \leftarrow \boldsymbol{\Phi}_{\text{MM}} \boldsymbol{\Theta} \boldsymbol{\Phi}_{\text{MM}}^\top \mathbf{X}$
8: **return** $\widetilde{\mathbf{X}}$

**Algorithm 2** Meta-training MM-FGCN

1: **Input:** graph dataset $S$, MM-FGCN parameters $(\boldsymbol{\theta}, \mathbf{w}; \boldsymbol{\xi})$, the empirical loss $\mathcal{L}(\cdot, \cdot)$. Learning rates $\beta_1, \beta_2 > 0$.
2: **Output:** optimized MM-FGCN $(\boldsymbol{\theta}^*, \mathbf{w}^*; \boldsymbol{\xi}^*)$
3: Split dataset $S_{\text{meta}}, S_{\text{main}} \leftarrow S$
4: **for** $t$ in $[T]$ **do**
5:      $B \leftarrow \text{MiniBatch}(S_{\text{meta}})$
6:      $\boldsymbol{\xi}' \leftarrow \boldsymbol{\xi} - \beta_1 \nabla_{\boldsymbol{\xi}} \mathcal{L}_B(\boldsymbol{\theta}, \mathbf{w}; \boldsymbol{\xi})$
7:      $B' \leftarrow \text{MiniBatch}(S_{\text{main}})$
8:      $(\boldsymbol{\theta}, \mathbf{w}; \boldsymbol{\xi}) \leftarrow (\boldsymbol{\theta}, \mathbf{w}; \boldsymbol{\xi}) - \beta_2 \nabla \mathcal{L}_{B'}(\boldsymbol{\theta}, \mathbf{w}; \boldsymbol{\xi}')$
9: **end for**
10: $(\boldsymbol{\theta}^*, \mathbf{w}^*; \boldsymbol{\xi}^*) \leftarrow (\boldsymbol{\theta}, \mathbf{w}; \boldsymbol{\xi})$
11: **return** $(\boldsymbol{\theta}^*, \mathbf{w}^*; \boldsymbol{\xi}^*)$

Figure 3: Left: the computation of MMFS-based multiresolution graph convolution operator. Right: implementation of MM-FGCN meta-training algorithm.

where $\text{MetaFGConv}_{\boldsymbol{\Theta}, \boldsymbol{\xi}}(\cdot)$ is the meta-framelet-based graph convolutional operator as defined in Algorithm 1, $h$ is a fixed classifier (e.g. softmax), $\mathbf{W}_l \in \mathbb{R}^{d_{l-1} \times d_l}$ are learnable weight matrices, and $\sigma(\cdot)$ is the activation function. We define $(\boldsymbol{\theta}, \mathbf{w})$ as the base-parameters, and define $\boldsymbol{\xi}$ as the meta-parameters. By design, the MM-FGCN is permutational invariant (Maron et al., 2019), and is equipped with a learnable multiresolution transform that adapts to each graph instance.

Following the optimization paradigm introduced in MAML (Finn et al., 2017; Hospedales et al., 2022), we employ meta-learning to train the MM-FGCN model. We aim to acquire a multiresolution transformation that enables the MM-FGCN backbone to adaptively and effectively represent individual graph instances. Specifically, our objective is

$$\min_{\boldsymbol{\theta}} \mathcal{L}_S(\boldsymbol{\theta}, \boldsymbol{\xi}^*(\boldsymbol{\theta})), \text{ s.t. } \boldsymbol{\xi}^*(\boldsymbol{\theta}) = \arg\min_{\boldsymbol{\xi}} \mathcal{L}(\boldsymbol{\theta}, \boldsymbol{\xi}), \tag{8}$$

$$\mathcal{L}_S(\boldsymbol{\theta}, \boldsymbol{\xi}) = \frac{1}{|S|} \sum_{(G,y) \in S} L(\text{MMFGCN}_{\boldsymbol{\theta}, \mathbf{w}; \boldsymbol{\xi}}(G), y), \tag{9}$$

where $L(\cdot, \cdot)$ is a loss function, e.g. the cross entropy. As outlined in Algorithm 2, we partition the training data into two sets: a meta-training set and a standard training set. In each iteration, we initiate an update to the meta-parameter $\boldsymbol{\xi}$, denoted as $\boldsymbol{\xi}'$, through gradient descent computed on a batch of meta-training data. Subsequently, we proceed to update all parameters $(\boldsymbol{\theta}, \mathbf{w}, \boldsymbol{\xi})$ using the full gradient evaluation at $(\boldsymbol{\theta}, \mathbf{w}, \boldsymbol{\xi}')$ based on the standard training data.

## 5 EXPERIMENTS

### 5.1 NODE CLASSIFICATION

**Datasets.** We conduct experiments on both assortative and disassortative graph datasets. A dataset is called assortative if its neighboring nodes usually have similar labels and features (Ma et al.,

Table 1: Test accuracy (in percentage) for citation networks with standard deviation after $\pm$. The results with the best performance are highlighted with *.

| Method | Assortative | | | Disassortative | | | | |
|---|---|---|---|---|---|---|---|---|
| | Cora | Citeseer | Pubmed | Cornell | Texas | Wisconsin | Chameleon | Squirrel |
| MLP | $55.1 \pm 1.1$ | $46.5 \pm 1.3$ | $71.4 \pm 0.7$ | $81.6 \pm 6.3$ | $81.3 \pm 7.1$ | $84.9 \pm 5.3$ | $48.5 \pm 3.0$ | $31.5 \pm 1.4$ |
| SPECTRAL (Bruna et al., 2014) | $73.3 \pm 1.4$ | $58.9 \pm 0.8$ | $73.9 \pm 0.6$ | $52.1 \pm 7.2$ | $57.5 \pm 7.5$ | $56.7 \pm 5.1$ | $62.4 \pm 2.4$ | $53.8 \pm 2.2$ |
| CHEBYSHEV (Defferrard et al., 2016a) | $81.2 \pm 1.2$ | $69.8 \pm 0.9$ | $74.4 \pm 0.6$ | $53.1 \pm 7.8$ | $59.1 \pm 7.2$ | $55.3 \pm 5.3$ | $60.2 \pm 2.9$ | $55.4 \pm 2.5$ |
| GWNN (Xu et al., 2019b) | $82.8 \pm 0.9$ | $71.7 \pm 1.1$ | $79.1 \pm 0.8$ | $56.8 \pm 7.6$ | $63.1 \pm 6.9$ | $61.2 \pm 4.9$ | $63.7 \pm 2.8$ | $55.4 \pm 2.3$ |
| MPNN Gilmer et al. (2017) | $78.0 \pm 1.1$ | $64.0 \pm 1.9$ | $75.6 \pm 1.0$ | $52.3 \pm 7.0$ | $58.2 \pm 5.4$ | $56.4 \pm 5.1$ | $60.8 \pm 2.7$ | $53.1 \pm 2.3$ |
| GRAPHSAGE Hamilton et al. (2017a) | $74.5 \pm 0.8$ | $67.2 \pm 1.0$ | $76.8 \pm 0.6$ | $54.2 \pm 7.8$ | $60.5 \pm 7.2$ | $58.7 \pm 5.3$ | $62.4 \pm 2.9$ | $55.4 \pm 2.5$ |
| LANCZOSNET Liao et al. (2019) | $79.5 \pm 1.8$ | $66.2 \pm 1.9$ | $78.3 \pm 0.3$ | $53.1 \pm 7.5$ | $60.4 \pm 7.2$ | $57.1 \pm 4.7$ | $65.2 \pm 2.5$ | $54.1 \pm 2.1$ |
| GCN Kipf & Welling (2016) | $81.5 \pm 1.2$ | $70.3 \pm 0.9$ | $79.0 \pm 0.4$ | $54.2 \pm 7.3$ | $61.1 \pm 7.0$ | $59.6 \pm 4.5$ | $67.6 \pm 2.4$ | $54.9 \pm 1.9$ |
| GAT (Veličković et al., 2018) | $83.0 \pm 0.7$ | $72.5 \pm 0.7$ | $79.0 \pm 0.3$ | $56.3 \pm 4.3$ | $57.9 \pm 6.1$ | $57.8 \pm 4.3$ | $65.0 \pm 3.7$ | $51.3 \pm 2.5$ |
| GIN+0 Xu et al. (2018) | $81.7 \pm 1.3$ | $71.4 \pm 0.8$ | $79.2 \pm 0.3$ | $57.4 \pm 7.8$ | $61.4 \pm 5.9$ | $58.3 \pm 7.2$ | $62.7 \pm 2.7$ | $38.0 \pm 1.8$ |
| GIN+$\epsilon$ Xu et al. (2018) | $81.6 \pm 1.4$ | $71.5 \pm 0.8$ | $79.1 \pm 0.4$ | $59.2 \pm 6.5$ | $60.5 \pm 6.2$ | $61.1 \pm 6.8$ | $61.4 \pm 2.2$ | $37.2 \pm 1.5$ |
| GraphGPS Rampášek et al. (2022) | $83.1 \pm 0.7$ | $72.3 \pm 0.8$ | $79.5 \pm 0.4$ | $67.2 \pm 7.7$ | $79.5 \pm 5.6$ | $76.9 \pm 4.9$ | $68.2 \pm 2.5$ | $58.4 \pm 1.6$ |
| NLMLP Liu et al. (2021a) | $68.5 \pm 1.9$ | $61.2 \pm 1.6$ | $71.8 \pm 0.9$ | $84.9 \pm 5.7$ | $85.4 \pm 3.8$ | $87.3 \pm 4.3$ | $50.7 \pm 2.2$ | $33.7 \pm 1.5$ |
| NLGCN Liu et al. (2021a) | $79.4 \pm 1.5$ | $70.2 \pm 1.4$ | $77.9 \pm 0.7$ | $57.6 \pm 5.5$ | $65.5 \pm 6.6$ | $60.2 \pm 5.3$ | $70.1 \pm 2.9$ | $59.0 \pm 1.2$ |
| NLGAT Liu et al. (2021a) | $80.1 \pm 1.3$ | $71.2 \pm 1.5$ | $78.1 \pm 0.7$ | $54.7 \pm 7.6$ | $62.6 \pm 7.1$ | $56.9 \pm 7.3$ | $65.7 \pm 1.4$ | $56.8 \pm 2.5$ |
| Geom-GCN-I Pei et al. (2020) | $80.0 \pm 1.2$ | $71.3 \pm 0.8$ | $78.2 \pm 0.5$ | $56.7 \pm 8.6$ | $57.5 \pm 5.8$ | $58.2 \pm 4.9$ | $60.3 \pm 2.7$ | $33.3 \pm 1.4$ |
| PyGNN Geng et al. (2023) | $83.3 \pm 0.9$ | $72.9 \pm 0.8$ | $79.8 \pm 0.4$ | $75.3 \pm 8.6$ | $79.2 \pm 4.6$ | $76.9 \pm 4.5$ | $65.4 \pm 2.5$ | $59.3 \pm 1.7$ |
| UFGCONV-S Zheng et al. (2021a) | $83.0 \pm 0.5$ | $71.0 \pm 0.6$ | $79.4 \pm 0.4$ | $67.8 \pm 8.0$ | $75.9 \pm 4.8$ | $72.4 \pm 4.2$ | $62.8 \pm 2.3$ | $57.6 \pm 1.5$ |
| UFGCONV-R Zheng et al. (2021a) | $83.6 \pm 0.6$ | $72.7 \pm 0.6$ | $79.9 \pm 0.1$ | $68.9 \pm 8.3$ | $77.2 \pm 4.7$ | $73.5 \pm 4.1$ | $63.1 \pm 2.4$ | $57.2 \pm 1.5$ |
| **MM-FGCN (Ours)** | $\mathbf{84.4^*} \pm 0.5$ | $\mathbf{73.9^*} \pm 0.6$ | $\mathbf{80.7^*} \pm 0.2$ | $\mathbf{88.9^*} \pm 8.3$ | $\mathbf{86.1^*} \pm 4.5$ | $\mathbf{88.5^*} \pm 4.1$ | $\mathbf{73.97^*} \pm 2.1$ | $\mathbf{67.5^*} \pm 1.2$ |

Table 2: Performance comparison for graph property prediction. QM7 is a regression task in MSE; others are for classification in test accuracy in percentage. The results with the best performance are highlighted with *.

| Pooling Operators | PROTEINS ($\uparrow$) | Mutagenicity ($\uparrow$) | D&D ($\uparrow$) | NCI1 ($\uparrow$) | Ogbg-molhiv ($\uparrow$) | QM7 ($\downarrow$) |
|---|---|---|---|---|---|---|
| TOPKPool | $73.48 \pm 3.57$ | $79.84 \pm 2.46$ | $74.87 \pm 4.12$ | $75.11 \pm 3.45$ | $78.14 \pm 0.62$ | $175.41 \pm 3.16$ |
| AttentionPool | $73.93 \pm 5.37$ | $80.25 \pm 2.22$ | $77.48 \pm 2.65$ | $74.04 \pm 1.27$ | $74.44 \pm 2.12$ | $177.99 \pm 2.22$ |
| SAGPool | $75.89 \pm 2.91$ | $79.86 \pm 2.36$ | $74.96 \pm 3.60$ | $76.30 \pm 1.53$ | $75.26 \pm 2.29$ | $41.93 \pm 1.14$ |
| SUMPool | $74.91 \pm 4.08$ | $80.69 \pm 3.26$ | $76.96 \pm 1.70$ | $76.96 \pm 1.70$ | $77.41 \pm 1.16$ | $42.09 \pm 0.91$ |
| MAX | $73.57 \pm 3.94$ | $78.83 \pm 1.70$ | $75.80 \pm 4.11$ | $75.96 \pm 1.82$ | $78.16 \pm 1.33$ | $177.48 \pm 4.70$ |
| MEAN | $73.13 \pm 3.18$ | $80.37 \pm 2.44$ | $76.89 \pm 2.23$ | $73.70 \pm 2.55$ | $78.21 \pm 0.90$ | $177.49 \pm 4.69$ |
| UFGPool-SUM | $77.77 \pm 2.60$ | $81.59 \pm 1.40$ | $80.92 \pm 1.68$ | $77.88 \pm 1.24$ | $78.80 \pm 0.56$ | $41.74 \pm 0.84$ |
| UFGPool-SPECTRUM | $77.23 \pm 2.40$ | $82.05 \pm 1.28$ | $79.83 \pm 1.88$ | $78.36 \pm 0.77$ | $78.36 \pm 0.77$ | $41.67 \pm 0.95$ |
| **MM-FGPool (Ours)** | $\mathbf{78.07^*} \pm 2.36$ | $\mathbf{83.91^*} \pm 1.32$ | $\mathbf{81.51^*} \pm 1.55$ | $\mathbf{78.57^*} \pm 0.82$ | $\mathbf{79.12^*} \pm 0.85$ | $\mathbf{41.19^*} \pm 0.88$ |

2022), as observed in citation networks and community networks. In contrast, disassortative datasets, such as co-occurrence networks and webpage linking networks, consist of numerous nodes with identical labels that are distant from one another. In this paper, we evaluate the performance of our MM-FGCN on assortative datasets, including Cora, Citeseer, and Pubmed (Sen et al., 2008), as well as disassortative datasets, including Cornell (Craven et al., 1998), Texas (Craven et al., 1998), Wisconsin (Craven et al., 1998), Chameleon (Rozemberczki et al., 2021), and Squirrel (Rozemberczki et al., 2021). For assortative datasets, following the configuration in (Kipf & Welling, 2016), we allocate 20 nodes per class for training, 1,000 nodes for testing, and 500 for validation. As for disassortative datasets, we divide each dataset into training, validation, and test sets using a split ratio of 60%:20%:20%. All experimental results are averaged over 10 independent repetitions.

**Baselines.** We benchmark MM-FGCN against various competitive baselines on node classification tasks, including MLP, CHEBYSHEV (Defferrard et al., 2016a), GCN (Kipf & Welling, 2016), SPECTRAL CNN (Bruna et al., 2014), GWNN (Xu et al., 2019b), MPNN (Gilmer et al., 2017), GRAPHSAGE (Hamilton et al., 2017a), LANCZOSNET (Liao et al., 2019), GAT (Veličković et al., 2018), Non-Local GNN (Liu et al., 2021a), Geom-GCN (Pei et al., 2020), two variants of UFGConv (Zheng et al., 2021a), i.e. UFGConvShrinkage and UFGConvRelu, and PyGNN (Geng et al., 2023). We adhere to the original implementations of the baseline models as described in their respective papers. As for MM-FGCN, the implementation details are elaborated in the Appendix A.

**Results.** As presented in Table 1, our proposed MM-FGCN model demonstrates state-of-the-art performance compared to all baseline models on both assortative and disassortative datasets. For disassortative datasets, compared to GCN, MM-FGCN achieves a significant performance gain of 34.7%, 25%, and 28.9% on the Cornel, Texas, and Wisconsin datasets, respectively. This evidence highlights that in disassortative datasets, where the node homophily is diminished and conventional models based on low-pass filters such as GCN struggle to capture effective graph representations. In contrast, MM-FGCN demonstrates its capability of learning a multiresolution framelet transform that dynamically adapts to the characteristics of each graph dataset. More experiments and results on node classifications are elaborated in the Appendix B.

Table 3: Ablation study on the meta-framelet learner and the meta-learning algorithm. Test accuracy (in percentage) with standard deviation after $\pm$. are reported.

| Methods | Graph Classification | | | Node Classification | | | | | |
|---|---|---|---|---|---|---|---|---|---|
| | Mutagenicity | D&D | NCI1 | Cora | Citeseer | Cornell | Texas | Chameleon | Squirrel |
| (a) Haar-type | $81.4 \pm 1.4$ | $80.9 \pm 1.7$ | $75.8 \pm 1.3$ | $83.3 \pm 0.5$ | $72.7 \pm 0.7$ | $77.8 \pm 7.9$ | $75.6 \pm 9.8$ | $54.6 \pm 6.6$ | $52.2 \pm 2.1$ |
| (b) Linear-type | $81.6 \pm 1.4$ | $80.6 \pm 1.8$ | $75.1 \pm 1.1$ | $83.0 \pm 0.6$ | $71.8 \pm 0.9$ | $76.1 \pm 7.5$ | $72.8 \pm 9.5$ | $54.3 \pm 2.1$ | $54.7 \pm 1.7$ |
| (c) Quadratic-type | $81.1 \pm 1.3$ | $80.3 \pm 1.9$ | $74.8 \pm 1.4$ | $82.7 \pm 0.7$ | $71.1 \pm 0.7$ | $76.7 \pm 8.9$ | $72.2 \pm 9.4$ | $57.5 \pm 2.7$ | $53.1 \pm 1.8$ |
| (d) Trainable framelet transforms | $82.3 \pm 1.4$ | $81.0 \pm 1.7$ | $75.9 \pm 0.9$ | $82.9 \pm 0.5$ | $72.2 \pm 0.7$ | $78.2 \pm 8.5$ | $77.9 \pm 9.2$ | $62.6 \pm 2.6$ | $59.2 \pm 2.2$ |
| (e) **MM-FGCN (Ours)** | $\mathbf{83.9^* \pm 1.3}$ | $\mathbf{81.5^* \pm 1.5}$ | $\mathbf{78.5^* \pm 0.8}$ | $\mathbf{84.4^* \pm 0.5}$ | $\mathbf{73.9^* \pm 0.6}$ | $\mathbf{88.9^* \pm 8.3}$ | $\mathbf{86.1^* \pm 4.5}$ | $\mathbf{73.9^* \pm 2.1}$ | $\mathbf{67.5^* \pm 1.2}$ |

## 5.2 GRAPH CLASSIFICATION

We assess the efficacy of MM-FGCN on 6 benchmark graph classification and regression datasets, including D&D (Dobson & Doig, 2003), PROTEINS (Dobson & Doig, 2003), NCI1 (Wale et al., 2008), Mutagenicity (Kazius et al., 2005), Ogbg-molhiv (Hu et al., 2020), and QM7 (Blum & Reymond, 2009). Following the configuration of Zheng et al. (2021a), each dataset is split into a training, validation, and test set by a ratio of 80%, 10%, and 10%. The results are averaged over 10 independent repetitions. We also compare MM-FGPool with graph classification methods based on the conventional GCN backbone together with various state-of-the-art pooling strategies, including SUM, MEAN, MAX pooling, TOPKPool (Gao & Ji, 2019), AttentionPool (Li et al., 2016), SAGPool (Lee et al., 2019), UFGPool-SUM, and UFGPool-SPECTRUM (Zheng et al., 2021a). The implementation details can be found in Appendix A. The results are shown in Table 2, and our model achieves the highest performance among all the baselines on the five datasets, demonstrating the effectiveness of MM-FGPool in aggregating graph information on various datasets.

## 5.3 ABLATION STUDIES

To validate the benefits of using a meta-framelet learner, in Table 3, we show the MM-FGCN variants with handcrafted filters (Dong, 2017b) (e.g. (a) Haar-type, (b) linear-type, (c) quadratic-type framelet filters). To assess the performance improvement achieved by the meta-learning algorithm elaborated in Algorithm 2, we compare it against a direct training scheme where both $\theta$ and $\xi$ are updated simultaneously, as shown in row (d) trainable framelet transforms of Table 3. According to the results, models with trainable meta-framelet generators outperform those with fixed and handcrafted graph transforms, highlighting the necessity of using trainable graph transforms for enhanced performance. Furthermore, using a meta-framelet learner indeed brings performance gain compared to using directly trainable filters, showing that the meta-framelet learner enhances the capacity of MM-FGCN. We also show that meta-learning contributes to improvement in the generalization performance of our MM-FGCN, leading to more discriminative graph representations. Extra ablation studies on the hyperparameters of MM-FGCN and visualizations are detailed in Appendix B.

## 6 CONCLUSION

In this paper, we present MM-FGCN, a spectral-based model for adaptive multiresolution representation learning for varying graph instances. Our MM-FGCN model is equipped with a set of trainable multiresolution bases, which can be simply and efficiently constructed based on a set of meta-band-pass filters. By optimizing the meta-filters, MM-FGCN learns an adaptive frequency partition of the graph spectrum domain, enabling us to perform a customized multiresolution transform on each graph instance. Comprehensive experiments show that our proposed method exhibits high performance and adaptivity to various types of graphs, including graph and node classification for dissortative and assortative graphs from various domains.

## ACKNOWLEDGEMENTS

This research is supported, in part, by Alibaba Group through the Alibaba Innovative Research (AIR) Program and Alibaba-NTU Singapore Joint Research Institute (JRI), Nanyang Technological University, Singapore. Tianze Luo wishes to extend his gratitude to the Alibaba Innovative Research (AIR) Program, NTU, Singapore, for their support. Sinno J. Pan thanks the support of the Hong Kong Jockey Club Charities Trust to the JC STEM Lab of Integration of Machine Learning and Symbolic Reasoning and the Microsoft Research Asia collaborative research grant.

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

# A  EXPERIMENT DETAILS

## A.1  DETAILS ON GRAPH DATASETS

We show the statistics of the datasets for node classification in Table 4. For each dataset, we list its graph statistics, data split, and homophily score, which is computed as follows.

**Homophily.** We adopt the homophily indicator $H(\mathcal{G})$ of the graph $\mathcal{G}$ from Liu et al. (2021b), which can be calculated as:

$$H(\mathcal{G}) = \frac{1}{|V|} \sum_{v \in V} \frac{| \{u : u \in \mathcal{N}(v) \text{ and } y(u) = y(v)\} |}{|\mathcal{N}(v)|},$$

where $| \{u : u \in \mathcal{N}(v) \text{ and } y(u) = y(v)\} |$ denotes the number of $v'$ s directly connected nodes who have the same label as $v$ and $|\mathcal{N}(v)|$ is the number of neighbouring nodes of $v$. Intuitively, high $H(\mathcal{G})$ indicates an assortative graph and vice versa.

Table 4: Statistics of the node-classification datasets used in our experiments. The homophily level of the dataset can be used to distinguish assortative and disassortative graph datasets.

| Datasets | Cora | Citeseer | Pubmed | Chameleon | Squirrel | Cornell | Texas | Wisconsin |
|---|---|---|---|---|---|---|---|---|
| Homophily | 0.83 | 0.71 | 0.79 | 0.25 | 0.22 | 0.11 | 0.06 | 0.16 |
| Splits | 140/500/1,000 | 120/500/1,000 | 60/500/1,000 | 60%/20%/20% | 60%/20%/20% | 60%/20%/20% | 60%/20%/20% | 60%/20%/20% |
| #Nodes | 2,708 | 3,327 | 19,717 | 2,277 | 5,201 | 183 | 183 | 251 |
| #Edges | 5,429 | 4,732 | 44,338 | 36,101 | 217,073 | 295 | 309 | 499 |
| #Features | 1,433 | 3,703 | 500 | 2,325 | 2,089 | 1,703 | 1,703 | 1,703 |
| #Classes | 7 | 6 | 3 | 5 | 5 | 5 | 5 | 5 |

For the graph classification task, we adopt Mutagenicity, D&D, NCI1, Ogbg-molhiv, and QM7 datasets. The D&D and PROTEINS datasets are used for protein structure classification, which aims to categorize proteins into enzyme and non-enzyme structures. The NCI1 dataset is used for identifying chemical compounds that inhibit lung cancer cells. The Mutagenicity dataset is used for recognizing mutagenic molecular compounds that have the potential for drug development. The QM7 dataset is used for predicting the atomization energy value of molecules. The Ogbg-Molhiv is a molecular property prediction dataset for predicting whether a molecule inhibits HIV virus replication or not. All the datasets contain more than 1,000 graphs with varying graph structures (in terms of the average number of nodes and edges, the average degree of nodes) and node features. The statistics of each dataset are displayed in Table 5.

Table 5: Summary of the datasets for the graph property prediction tasks.

| Datasets | PROTEINS | Mutagenicity | D&D | NCI1 | ogbg-molhiv | QM7 |
|---|---|---|---|---|---|---|
| # Graphs | 1,113 | 4,337 | 1,178 | 4,110 | 41,127 | 7,165 |
| Min # Nodes | 4 | 4 | 30 | 3 | 2 | 4 |
| Max # Nodes | 620 | 417 | 5,748 | 111 | 222 | 23 |
| Avg # Nodes | 39 | 30 | 284 | 30 | 26 | 15 |
| Avg # Edges | 73 | 31 | 716 | 32 | 28 | 123 |
| # Features | 3 | 14 | 89 | 37 | 9 | 0 |
| # Classes | 2 | 2 | 2 | 2 | 2 | 1 (R) |

## A.2  IMPLEMENTATION DETAILS

**Hyper-parameters**

We implement our model using PyTorch. We set the default number of filters as four, which is suitable for most of the datasets. The default Chebyshev approximation order is set to 6. The dimension of hidden variables is searched from $\{16, 32, 64\}$, and the level of filters are selected from $\{2, 3, 4, 5\}$. Other hyperparameters are set at: 0.001 for the learning rate, 0.001 for weight decay, 0.5 for dropout, and 2 for the number of MM-FGConv layers. These hyper-parameters are used for both node and graph classification tasks. For the graph classification task, we further apply our proposed MM-FGPool operation as elaborated in Section 4.3, followed by a linear classifier. For baseline methods in node classification, we adopt the code from the author's released implementation with the default settings. For graph classification, we adopt the experiment setting form Zheng et al.

(2021a), where we use two-layer GCN networks followed by the pooling methods listed in Table 2. Specifically, this experiment setting is comparable to our model's design, where we also adopt two convolutional layers and one pooling layer.

**Design of the neural network $\mathcal{M}_\xi$**

In practice, the neural network $\mathcal{M}_\xi$ is set to be a 2-layer standard GCN network that maps a graph data $(\mathbf{X}, \mathbf{A}) \in \mathbb{R}^{n \times d} \times \mathbb{R}^{n \times n}$ into a $2(I-1)$-dimensional vector $\omega \in \mathbb{R}^{2(I-1)}$, i.e. the parameters of the meta-filters $\{g_{r,i}^{\boldsymbol{\omega}}\}_{r,i}$.

Specifically, we set $\mathcal{M}_\xi(\mathbf{X}, \mathbf{A}) = 1/n \cdot 1^\top \sigma(\mathbf{L}\,\sigma(\mathbf{L}\mathbf{X}\mathbf{W}_1)\mathbf{W}_2)\mathbf{W}_3$, where $\mathbf{W}_1 \in \mathbb{R}^{d \times d/2}$, $\mathbf{W}_2 \in \mathbb{R}^{d/2 \times d/2}$ and $\mathbf{W}_3 \in \mathbb{R}^{d/2 \times 2(I-1)}$ are trainable weights, $\mathbf{L}$ is the normalized graph Laplacian matrix computed from the adjacency matrix $\mathbf{A}$, and $\sigma(\cdot)$ is the ReLU activation function. We denote the collection of all the trainable weights as $\xi = (\mathbf{W}_1, \mathbf{W}_2, \mathbf{W}_3)$.

After obtaining the $2(I-1)$-dimensional output $\boldsymbol{\omega} \in \mathbb{R}^{2(I-1)}$ from $\mathcal{M}_{\boldsymbol{\xi}}$, we use Equation (3) and Equation (4) to determine the exact formulation of the meta-filters $\{g_{r,i}^{\boldsymbol{\omega}}\}_{r,i}$. To determine the formulation of $(r,i)$-th meta-filter, we first decode its lower- and the upper-cutoff positions and margins, i.e. $c_i$, $c_{i+1}$, $\epsilon_i$, and $\epsilon_{i+1}$, from $\boldsymbol{\omega}$ using Equation (3). Then, the formulation of $g_{1,i}^{\boldsymbol{\omega}}$ (the bandpass starting from $c_i + \epsilon_i$ to $c_{i+1} + \epsilon_{i+1}$) can be directly derived from Equation (4). Finally, starting from $g_{1,i}^{\boldsymbol{\omega}}$, we can determine $g_{r,i}^{\boldsymbol{\omega}}$ in an iterative manner, using the second condition in Proposition 2. The visualization example of the meta-filters is shown in Figure 1 and Figure 2 of the manuscript.

**Split of $S_{\text{meta}}$ and $S_{\text{main}}$**

The $S_{\text{meta}}$ and $S_{\text{main}}$ are the data randomly split from the training data. In our experiment, we take 80% of the training data as the $S_{\text{main}}$ and 20% as $S_{\text{meta}}$. In practice, our MM-FGCN shares the same train and test dataset splitting scheme as other baseline methods. We first train the MM-FGCN by meta-training (Algorithm 2) on the training data. Then, we fix the trained parameters $\boldsymbol{\theta}, \boldsymbol{\xi}$, and we evaluate the accuracy of the MM-FGCN on the testing dataset. The dataset splitting in line 3 of Algorithm 2 is done randomly within the training data. Hence, in our experiments, the MM-FGCN is evaluated under the same data resource as other baselines.

## B  EXTRA EXPERIMENTS

### B.1  EXPERIMENTS ON RANDOM 60%/20%/20% SPLITS

The main results of the full sets of node classification experiments with statistics of datasets are summarized in Table 1, and Table 4. For a fair comparison with the state-of-the-art methods, we list the additional experiments for node classification on homophily graph datasets with 60%/20%/20% split. Corresponding results are shown in Table 6.

|  | Cora | Citeseer | Pubmed |
|---|---|---|---|
| MLP | $76.44 \pm 0.30$ | $76.25 \pm 0.28$ | $86.43 \pm 0.13$ |
| GCN | $87.78 \pm 0.96$ | $81.39 \pm 1.23$ | $88.9 \pm 0.32$ |
| GAT | $76.70 \pm 0.42$ | $67.20 \pm 0.46$ | $83.28 \pm 0.12$ |
| GraphSAGE | $86.58 \pm 0.26$ | $78.24 \pm 0.30$ | $86.85 \pm 0.11$ |
| Geom-GCN | $85.27 \pm 1.28$ | $77.99 \pm 0.92$ | $90.05 \pm 0.17$ |
| ACM-GCN | $88.62 \pm 1.22$ | $81.68 \pm 0.97$ | $90.66 \pm 0.47$ |
| GCNII | $88.98 \pm 1.33$ | $81.58 \pm 1.3$ | $89.8 \pm 0.30$ |
| MM-FGCN (Ours) | $\mathbf{89.89}^* \pm 1.12$ | $\mathbf{82.97}^* \pm 0.85$ | $\mathbf{91.52}^* \pm 0.21$ |

Table 6: Test accuracy for classifications on homophily graphs under 60%/20%/20% random split.

### B.2  EXPERIMENTS ON FIXED 48%/32%/20% SPLITS

We further conduct the node classification experiments on data split with fixed 48%/32%/20% according to Pei et al. (2020). The corresponding results are shown in Table 7.

| | Cornell | Wisconsin | Texas | Chameleon | Squirrel | Cora | Citeseer | Pubmed |
|---|---|---|---|---|---|---|---|---|
| Geom-GCN | $60.54 \pm 3.67$ | $64.51 \pm 3.66$ | $66.76 \pm 2.72$ | $60.00 \pm 2.81$ | $38.15 \pm 0.92$ | $85.35 \pm 1.57$ | $78.02 \pm 1.15$ | $89.95 \pm 0.47$ |
| GGCN | $85.68 \pm 6.63$ | $86.86 \pm 3.29$ | $84.86 \pm 4.55$ | $71.14 \pm 1.84$ | $55.17 \pm 1.58$ | $87.95 \pm 1.05$ | $77.14 \pm 1.45$ | $89.15 \pm 0.37$ |
| H2GCN | $82.70 \pm 5.28$ | $87.65 \pm 4.98$ | $84.86 \pm 7.23$ | $60.11 \pm 2.15$ | $36.48 \pm 1.86$ | $87.87 \pm 1.20$ | $77.11 \pm 1.57$ | $89.49 \pm 0.38$ |
| MixHop | $73.51 \pm 6.34$ | $75.88 \pm 4.90$ | $77.84 \pm 7.73$ | $60.50 \pm 2.53$ | $43.80 \pm 1.48$ | $87.61 \pm 0.85$ | $76.26 \pm 1.33$ | $85.31 \pm 0.61$ |
| Geom-GCN | $60.54 \pm 3.67$ | $64.51 \pm 3.66$ | $66.76 \pm 2.72$ | $60.00 \pm 2.81$ | $38.15 \pm 0.92$ | $85.35 \pm 1.57$ | $78.02 \pm 1.15$ | $89.95 \pm 0.47$ |
| ACM-GCN | $85.14 \pm 6.07$ | $88.43 \pm 3.22$ | $87.84 \pm 4.4$ | $69.14 \pm 1.91$ | $55.19 \pm 1.49$ | $87.91 \pm 0.95$ | $77.32 \pm 1.7$ | $90.00 \pm 0.52$ |
| GCNII | $77.86 \pm 3.79$ | $80.39 \pm 3.40$ | $77.57 \pm 3.83$ | $63.86 \pm 3.04$ | $38.47 \pm 1.58$ | $88.37 \pm 1.25$ | $77.33 \pm 1.48$ | $90.15 \pm 0.43$ |
| NLMLP | $84.9 \pm 5.7$ | $87.3 \pm 4.3$ | $85.4 \pm 3.8$ | $50.7 \pm 2.2$ | $33.7 \pm 1.5$ | $76.9 \pm 1.8$ | $73.4 \pm 1.9$ | $88.2 \pm 0.5$ |
| NLGCN | $57.6 \pm 5.5$ | $60.2 \pm 5.3$ | $65.5 \pm 6.6$ | $70.1 \pm 2.9$ | $59.0 \pm 1.2$ | $88.1 \pm 1.0$ | $75.2 \pm 1.4$ | $89.0 \pm 0.5$ |
| NLGAT | $54.7 \pm 7.6$ | $56.9 \pm 7.3$ | $62.6 \pm 7.1$ | $65.7 \pm 1.4$ | $56.8 \pm 2.5$ | $88.5 \pm 1.8$ | $76.2 \pm 1.6$ | $88.2 \pm 0.3$ |
| **MM-FGCN (Ours)** | $\mathbf{87.35}^* \pm 6.18$ | $\mathbf{89.02}^* \pm 5.41$ | $\mathbf{89.31}^* \pm 1.56$ | $\mathbf{72.61}^* \pm 1.84$ | $\mathbf{61.34}^* \pm 1.22$ | $\mathbf{89.35}^* \pm 1.15$ | $79.86 \pm 1.42$ | $\mathbf{91.42}^* \pm 0.41$ |

Table 7: Test accuracy for node classification results under fixed 48%/32%/20% split.

### B.3 PERTURBATION RESILIENCE OF MM-FGCN.

In this study, we add extra experiments to assess the perturbation resilience of our MM-FGCN against noise perturbation present in input graph data, which is ubiquitous in real-world datasets. Particularly, we train the MM-FGCN with corrupted data that are contaminated by random noise of various magnitudes. The noise magnitude is controlled by the noise ratios, which are defined as the amount of randomly deleted edges (or randomly flipped binary-valued features) divided by the number of untainted edges (or features). We then investigate how the performance of the resultant models varies when the noise level change from 0 to 1. As illustrated in Figure 4, our MM-FGCN consistently outperforms the baselines with a remarkable margin even under the presence of considerable noise. Thus, the MM-FGCN demonstrates a strong noise resilience making it a highly promising solution for real-world applications.

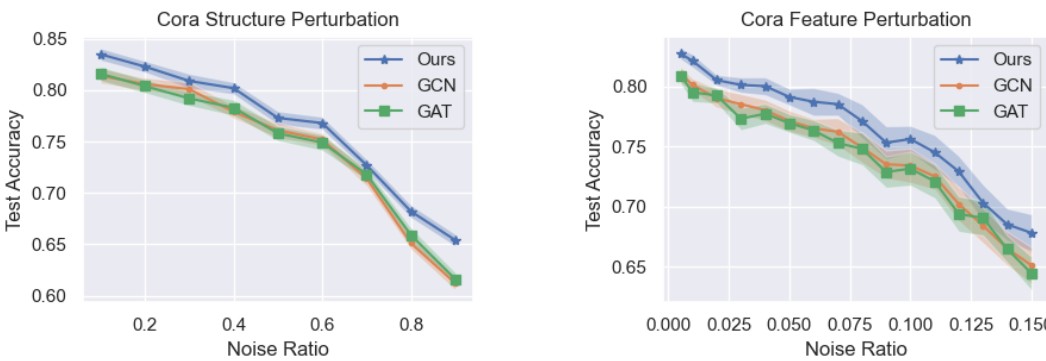

Figure 4: Noise resilience experiments with the edge (left) and feature (right) noise perturbations on Cora.

### B.4 ABLATION ON THE NUMBER OF THE META-FRAMELET GENERATORS

We analyze how the size of the meta-framelet generator (i.e. a set of spectral filters), $I$, affects the performance of MM-FGCN. With an insufficient amount of meta-filters, the model may fail to learn the optimal frequency partition and cannot disentangle graph signals into desirable frequency components. Intuitively, an overly small $I$ may hinder the learning of discriminative graph representations. Conversely, a large $I$ improves the precision of frequency partition learning but also increases computational expenses. This requires us to strike a balance between the by selectively choosing the value of $I$.

We evaluate the performance of MM-FGCN with different choices of $I$ over Cora, Citeseer, Chameleon, Squirrel, and D&D datasets. As shown in Figure 5 (a), the Meta-FCGN is able to achieve state-of-the-art performance by constructing the meta-framelet generator with only 3 filters. Overall, the model performance is stable and robust across different choices of $I$. In general, we recommend setting $I = 4$ for effective and efficient implementation.

### B.5 ABLATION ON THE ORDER OF CHEBYSHEV APPROXIMATION

Recall that the Chebyshev approximation trick Defferrard et al. (2016b) is applied in Algorithm 1 for efficient computation of each $g_{r,i}^{\omega}(\mathbf{L})$. Broadly speaking, using a higher-order Chebyshev approximation leads to a smaller approximation error w.r.t the meta-generator, but creates a greater computational overhead. As illustrated in Figure 5 (b), the performance of MM-FGCN is robust to Chebyshev approximation of different orders. Empirically, the MM-FGCN achieves optimal performance with a Chebyshev approximation of an order greater than 4. In contrast, a low-order Chebyshev approximation order incurs an undesirable approximation error, which hinders graph representation learning and impairs model generalization. We recommend using a higher than 4-order Chebyshev approximation for good model performance.

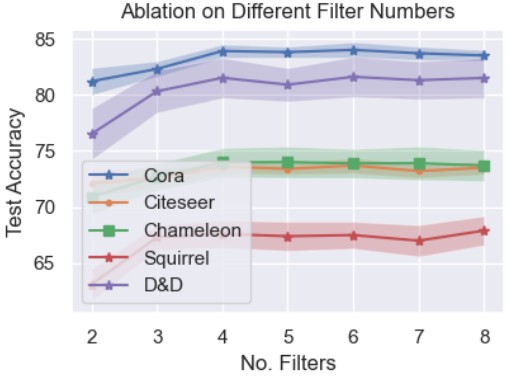
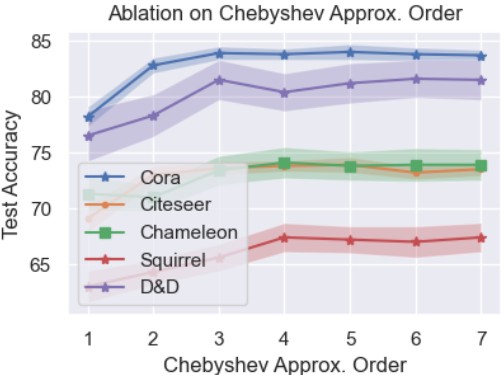

(a) Ablation on the number of filters used in the meta-framelet generator.

(b) Ablation on the order of Chebyshev approximation.

Figure 5: Ablation studies on MM-FGCN's hyperparameters.

### B.6 VISUALIZATION OF MM-FGCN REPRESENTATION

In this section, we empirically show that the MM-FGCN is able to produce more discriminative graph representations than conventional GCN, on both assortative (e.g. Cora) and disassortative (e.g. Cornell) datasets. In order to assess the quality of the learned graph representation, we visualize the hidden features generated by the penultimate layer of both MM-FGCN and GCN via t-SNE van der Maaten & Hinton (2008). As shown in Figure 6, the graph representation of our MM-FGCN is more spatially clustering than GCN, especially for the disassortative dataset that is more challenging for classification. In fact, the learned feature of MM-FGCN is strongly correlated to the label, which significantly facilitates node and graph classification tasks. In summary, the visualization of the learned hidden features validates that the adaptiveness and expressiveness of our MM-FGCN are beneficial to learning discriminative graph representations.

We also demonstrate the effectiveness of the MM-FGCN by visualising the filters learned by MM-FGCN in Figure 6. We can observe that the filters learned from the Cora dataset (assortative) more concentrate on the low-frequency signals than the filters learned from the Cornell dataset (disassortative). Due to the assortative properties, aggregating local information with low-frequency signals can benefit the model's performance on the Cora dataset. In contrast, as elaborated in Bo et al. (2021), high-frequency signals are useful for disassortative networks, which corresponds to our learned multi-resolution filters since more filters concentrate on the high-frequency part, providing a comprehensive feature extraction on the high-frequency signal. This phenomenon shows the adaptivity of our MM-FGCN on different types of graphs.

### B.7 ROBUSTNESS ANALYSIS

We show that our MM-FGCN inherently possesses greater robustness compared to conventional GCN models, even without relying on any specific robust data augmentation techniques in Appendix B.4. Furthermore, we observe that the robustness of our MM-FGCN can be further enhanced when combined with data augmentation methods. We assess the robustness of our MM-FGCN in the face

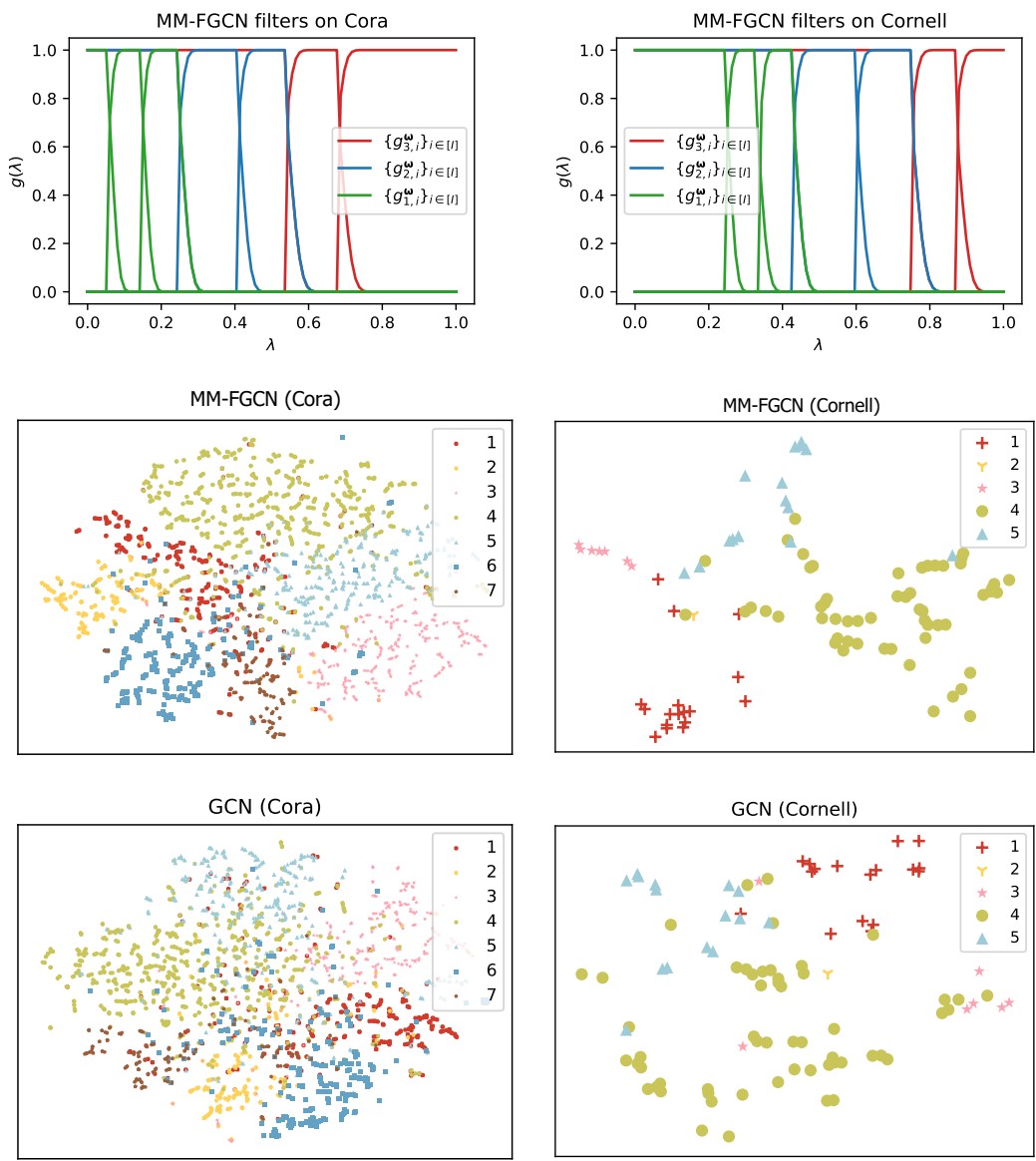

Figure 6: Meta-framelet generator (row 1) and feature visualization (row 2-3) on the test and validation sets of Cora (left, assortative) and Cornell (right, disassortative) datasets using MM-FGCN and GCN.

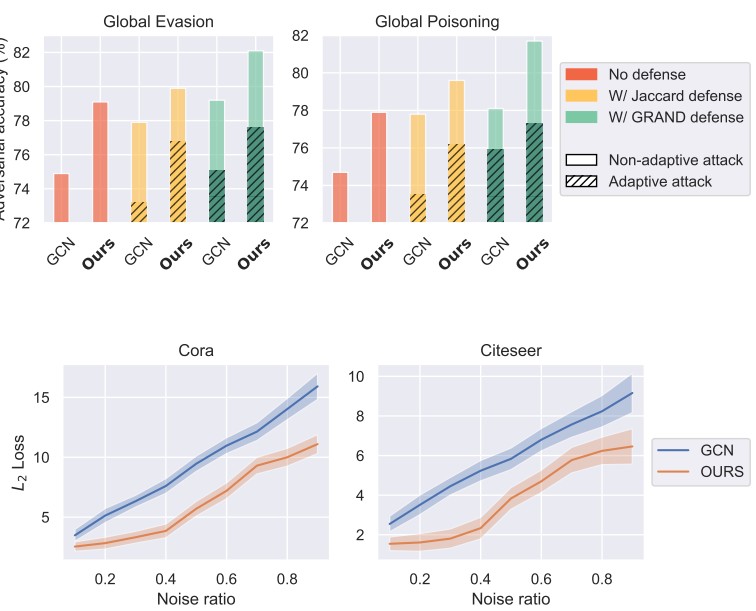

Figure 7: Top: the adversarial accuracy of MM-FCGN under graph attack. Bottom: representation distortion of each perturbation level.

of both adaptive and non-adaptive graph attacks Mujkanovic et al. (2022). These attacks encompass graph poisoning, which involves perturbing the adjacency matrix before training, and graph evasion, which perturbs the adjacency matrix after training. We not only compare our method with GCN but also evaluate two prevalent graph defense techniques, namely GRAND defense and Jaccard defense. We present the results on Cora in the top row of Figure 7, demonstrating that our MM-FGCN showcases substantially higher adversarial accuracy than GCN when subjected to graph attacks. Moreover, by incorporating GRAND and Jaccard defense techniques, the adversarial accuracy of MM-FGCN can be further enhanced. This highlights that the inherent robustness gain of MM-FGCN is independent of the benefits offered by graph defense methods, demonstrating our orthogonal contributions.

We also validate our MM-FGCN's immunity against realist graph topology perturbations. To evaluate the robustness of a well-trained model, we introduce noise of varying magnitudes into the adjacency matrix. Subsequently, we assess the deviation between the perturbed hidden representation and the original one using the $\ell_2$ distance metric. As depicted in the bottom row of Figure 7, our MM-FGCN demonstrates a distinct 'potential well', where the potential refers to the distance from the perturbed feature to the original, unperturbed feature. The results in Figure 7 show that our MM-FGCN has the capability to learn robust representations that remain intact by minor perturbations in the graph adjacency matrix. In contrast, conventional GCN experiences a progressively linear increase in representation distortion as the magnitude of perturbation grows.

## B.8 ABLATION STUDIES ON META-LEARNING

We compare the performance of MM-FGCN, both with and without meta-training, on the node classification datasets including Cora, Citeseer, PubMed, Cornell, Texas, Wisconsin, Chameleon, and Squirrel; and graph classification datasets including PROTEINS, Mutagenicity, D&D, NCI1, Ogbg-molhiv, and QM7. We show the ablation study results in Table 8 and Table 9.

| | Cora | Citeseer | PubMed | Cornell | Texas | Wisconsin | Chameleon | Squirrel |
|---|---|---|---|---|---|---|---|---|
| MM-FGCN w/o meta-learning | 82.9 ± 0.5 | 72.2 ± 0.7 | 80.1 ± 0.2 | 78.2 ± 8.5 | 77.9 ± 9.2 | 82.3 ± 4.8 | 62.6 ± 2.6 | 59.2 ± 2.2 |
| MM-FGCN | **84.4 ± 0.5** | **73.9 ± 0.6** | **80.7 ± 0.2** | **88.9 ± 8.3** | **86.1 ± 4.5** | **88.5 ± 4.1** | **73.97 ± 2.1** | **67.5 ± 1.2** |

Table 8: Comparison of with and without meta-learning framework on the node classification tasks.

|  | PROTEINS (↑) | Mutagenicity (↑) | D&D (↑) | NCI1 (↑) | Ogbg-molhiv (↑) | QM7 (↓) |
|---|---|---|---|---|---|---|
| MM-FGPool w/o meta-learning | 77.86 ± 2.53 | 82.3 ± 1.4 | 81.0 ± 1.7 | 75.9 ± 0.9 | 78.57 ± 0.73 | 41.65 ± 0.91 |
| MM-FGPool | **78.07 ± 2.36** | **83.91 ± 1.32** | **81.51 ± 1.55** | **78.57 ± 0.82** | **79.12 ± 0.85** | **41.19 ± 0.88** |

Table 9: Comparison of with and without meta-learning framework on the graph classification tasks.

As shown in Table 8 and Table 9, in practice, using meta-training (Algorithm 2) leads to a consistent performance gain on varying datasets in both graph and node classification settings. Moreover, as shown in Figure 8, the meta-training process accelerates the convergence of MM-FGCN and improves the generalization performance of MM-FGCN.

In intuition, using different data batches in the meta-step and main-step of Algorithm 2 helps MM-FGCN to avoid overfitting. In the meta-step, the meta-framelet learner $\xi$ is first optimized to improve the meta-framelet for the current backbone model $\theta$. Based on this optimized meta-framelet representation, the whole model is then evaluated and optimized with another newly sampled data batch. Using two identically distributed but distinct data batches, the $\xi$-meta-framelet transform will be cautiously updated until achieving good performance on both the meta- and main-data batches. Consequently, we anticipate that the meta-training process encourages MM-FGCN to find meta-framelet representations that can be generalized to different batches drawn from the underlying distribution, rather than merely following the gradient calculated on the current data batch, thus reducing the risk of overfitting.

**How does meta-learning work?**

Following Hospedales et al. (2022), in this paper, Meta-learning is understood as 'learning-to-learn', which refers to the process of improving the base learning algorithm over multiple learning episodes, i.e. the base- and meta-learning episodes. During base-learning, a base-learning algorithm solves the base task, such as graph/node classification. During meta-learning, a meta-algorithm updates the base learning algorithm such that the model learns to improve the meta objective.

In this paper, the base objective refers to 'learning a graph/node classifier using features based on a framelet transform parameterized by $\omega$', as shown in Equation (8) and Equation (9). We aim to design a meta-objective that refers to 'finding a good $\omega$ such that the $\omega$-base-objective can be achieved well'. To this end, as shown in the second term of Equation (8), we set the meta-objective to be 'finding the $\omega$ with the lowest classification loss, for any given network backbone parameter $\theta$'. The intuition behind this is that for any model backbone, we always want to choose the most compatible $\omega$-framelet transform, such that the model backbone $\theta$ can fully release its power, achieving a classification loss. By combining the base- and meta-objectives, we establish the meta-learning training paradigm of MM-FGCN as a bi-level optimization problem in Equation (8).

Unlike meta-learning models, where the base-learning algorithm is continuously improved during the meta-learning episode, a standard machine learning model is trained on a specific task using fixed, handcrafted base-learning algorithms. For instance, standard GCNs are trained to predict graph labels by minimizing the classification loss, using human-designed features like the graph Fourier transform. On the contrary, our MM-FGCN is able to learn suitable meta-framelet transform that is adaptive to varying graph instances and distributions via the meta-learning paradigm.

In a nutshell, using meta-learning improves the performance MM-FGCN on various graph learning scenarios. In intuition, meta-learning encourages MM-FGCN to learn meta-framelet representations that generalize well.

B.9 EXPERIMENTS ON LARGE-SCALE GRAPH DATASETS

Our MM-FGCN can be applied to large graphs, e.g. OGBN datasets, and it achieves a high performance. We conduct experiments on the OGBN-arxiv and OGBN-product datasets, and the results are listed in the following table.

To train our MM-FGCN, for OGBN-arxiv, we use the 128-dimensional feature vector as the node feature of each node; for OGBN-product, we use the 100-dimensional feature vector as each node's feature. It is worth noting that some state-of-the-art methods (such as SimTeG+TAPE+RevGAT, TAPE+RevGAT, and SimTeG+TAPE+GraphSAGE) adopt the raw title and abstract form the OGBN-arxiv and detailed product descriptions from OGBN-product as the node features, and achieve better

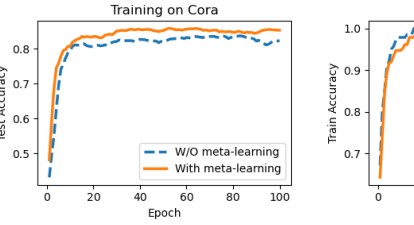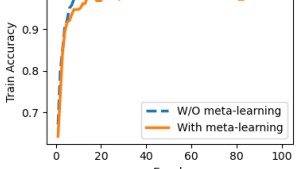

(a) Accuracy on the test set    (b) Accuracy on the train set

Figure 8: Comparison of MM-FGCN with and w/o meta-training on the Cora dataset. We report the train and test accuracy of each training step.

results due to using a good NLP module in extracting node features. Since we mainly compare the performance of the GNN module, we neglect the gain from using NLP modules such as GIANT-XRT or TAPE, etc. Instead, we directly use the preprocessed feature vectors provided in both datasets on our model and our baseline models.

**Training details**

To train the OGBN-Arxiv, we use the full-batch training. In the OGBN-products dataset, we use neighbor sampling with size = 10 (similar to UniMP [10]) to sample the subgraph during training. Other hyperparameter settings follow our experiments on other datasets as elaborated in the Appendix. We show the corresponding results in the following tables.

| Method | Test Acc. | Val. Acc. |
|---|---|---|
| MLP | $55.50 \pm 0.23$ | $57.65 \pm 0.12$ |
| NODE2VEC | $70.07 \pm 0.13$ | $71.29 \pm 0.13$ |
| GRAPHZOOM | $71.18 \pm 0.18$ | $72.20 \pm 0.07$ |
| P&L + C&S | $71.26 \pm 0.01$ | $73.00 \pm 0.01$ |
| GRAPHSAGE | $71.49 \pm 0.27$ | $72.77 \pm 0.17$ |
| GCN | $71.74 \pm 0.29$ | $73.00 \pm 0.17$ |
| DEEPERGCN | $71.92 \pm 0.17$ | $72.62 \pm 0.14$ |
| SIGN | $71.95 \pm 0.11$ | $73.23 \pm 0.06$ |
| GAAN | $71.97 \pm 0.18$ | – |
| UFGCONV-R | $71.97 \pm 0.12$ | $73.21 \pm 0.05$ |
| UniMP | $73.11 \pm 0.21$ | $74.50 \pm 0.05$ |
| Exphormer | $72.44 \pm 0.28$ | – |
| DRGAT | $74.16 \pm 0.07$ | $75.34 \pm 0.02$ |
| Ours | $\mathbf{74.25 \pm 0.15}$ | $\mathbf{75.46 \pm 0.08}$ |

Table 10: Experiment on the OGBN-Arxiv dataset.

| Method | Test Acc. | Val. Acc. |
|---|---|---|
| MLP | $55.50 \pm 0.23$ | $57.65 \pm 0.12$ |
| GCN-Cluster | $78.97 \pm 0.36$ | $92.12 \pm 0.09$ |
| GAT-Cluster | $79.23 \pm 0.78$ | $89.85 \pm 0.22$ |
| GAT-NeighborSampling | $79.45 \pm 0.59$ | – |
| GraphSAINT | $80.27 \pm 0.26$ | – |
| DeeperGCN | $80.90 \pm 0.20$ | $92.38 \pm 0.09$ |
| UniMP | $82.56 \pm 0.31$ | $93.08 \pm 0.17$ |
| AGDN | $83.34 \pm 0.27$ | $92.29 \pm 0.10$ |
| Ours | $\mathbf{84.03 \pm 0.23}$ | $\mathbf{93.57 \pm 0.12}$ |

Table 11: Experiment on the OGBN-Products dataset.

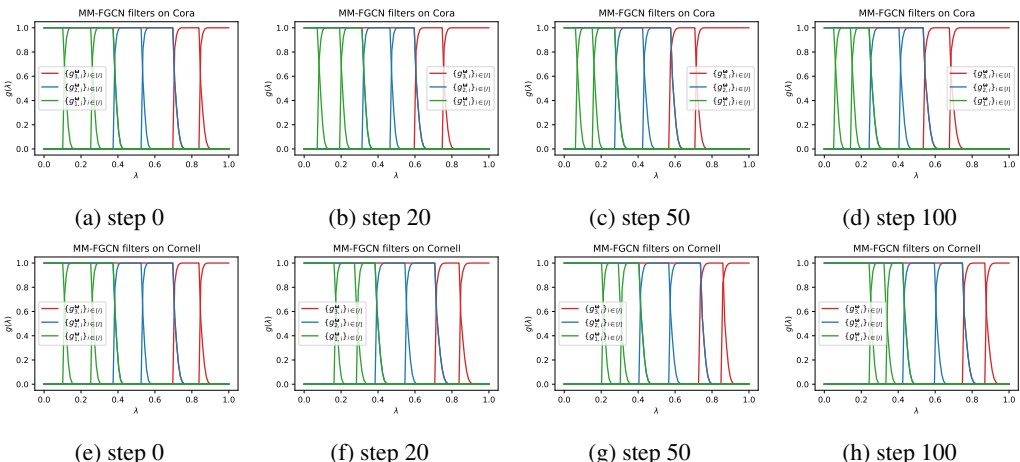

Figure 9: From (a) to (d): Variation of filters during the training process on the Cora dataset. From (e) to (h): Variation of filters during the training process on the Cornell dataset.

## B.10 VARIATION OF FILTERS DURING THE TRAINING PROCESS

We visualize the variation of filters during the training process on the Cora and Cornell datasets. We train our model on both datasets for 100 epochs and visualize the filters at training steps 0, 20, 50, and 100.

Recall that the Cora dataset contains a graph of a citation network, where each node represents a scientific publication and the node labels are the research domain. Thus, the node label (i.e. the graph signal of interest) varies smoothly across the nodes, emphasizing the importance of low-frequency information in learning good graph representations.

As shown in Figure 9 (a, b, c, d), we observe that MM-FGCN automatically learns to focus more on establishing the low-frequency representations: at the initial stage of the training process, the meta-filters are evenly separated and do not show special preference to the graph signals at specific frequencies. During the training process, the filters tend to concentrate on the low-frequency part of the graph signal to retrieve more refined low-frequency information of the graph.

The Cornell dataset contains a website network, where each node represents a web page that is manually classified into five categories: student, project, course, staff, and faculty. Edges represent hyperlinks between them. In this case, nodes with the same labels are not likely to link to each other. Thus, the node label (i.e. the graph signal of interest) varies drastically across neighbouring nodes, showing that high-frequency information is essential in learning good graph representations.

As shown in Figure 9 (e, f, g, h), we observe that MM-FGCN automatically learns to focus more on establishing the high-frequency representations: at the initial stage of the training process, the meta-filters are evenly separated and do not show special preference to the graph signals at specific frequencies. During the training process, the filters tend to concentrate on the high-frequency part of the graph signal to refine the high-frequency representations of the graph.

## B.11 TIME COMPLEXITY ANALYSIS

The complexity of a single forward pass of the meta-framelet convolution is approximately $r$ times as the cost of standard graph convolution, where $r$ is the number of meta-framelets. In practice, our MM-FGCN can be computed efficiently, without significant overhead.

Suppose we instantiate the meta-learner $\mathcal{M}_{\boldsymbol{\xi}}$ as a $L$-layer GCN, we use a $k$-order Chebyshev approximation for the $n \times n$ graph Laplacian matrix $\mathbf{L}$, and we set the number of filters as $r$. Then, for each $d$-dimensional input graph feature $\mathbf{H} \in \mathbb{R}^{n \times d}$, the computational complexity of meta-framelet convolution is $O(rn \times n \times d) + O(n \times n \times d) + O(r \times k^2 \times n^2) = O(rn^2(d + k^2))$. In this setting, the standard graph convolutional layer costs $O(n^2 d)$.

**Experiments on training and inference time cost**

In the following Table 12 and Table 13, we compare both the training and inference time cost of our MM-FGCN against training other baselines, including GCN, GAT, GraphSage and SVD-GCN. We report the time cost of one training epoch in Table 12, and we evaluate the time cost of on single forward pass in Table 13. Specifically, each training epoch of MM-FGCN includes both the base- and the meta-update step. And we set the Chebyshev approximation order $k$ and the number of meta-framelets $r$ as $k = 4$ and $r = 4$. We conduct the experiments on varying GNN architectures, with varying parameter budgets (e.g. model size). The experiments are conducted on a single 40G A100 GPU.

| Model | # Parameters | Acc. ($\uparrow$) | Time/Epoch ($\downarrow$) | # Parameters | Acc. ($\uparrow$) | Time/Epoch ($\downarrow$) | # Parameters | Acc. ($\uparrow$) | Time/Epoch ($\downarrow$) |
|---|---|---|---|---|---|---|---|---|---|
| GCN | 10,246 | 80.4 | 0.012 | 51,238 | 81.2 | 0.016 | 100,204 | 82.0 | 0.018 |
| GAT | 10,124 | 79.6 | 0.021 | 50,422 | 81.7 | 0.022 | 101,684 | 82.5 | 0.024 |
| GraphSage [4] | 10,076 | 78.9 | 0.248 | 51,682 | 80.9 | 0.281 | 107,216 | 81.9 | 0.315 |
| SVD-GCN [15] | 10,578 | 79.6 | 0.236 | 53,412 | 81.4 | 0.254 | 106,452 | 82.5 | 0.261 |
| MM-FGCN (ours) w/o meta-training | 10,674 | 82.2 | 0.048 | 52,546 | 83.7 | 0.050 | 101,046 | 84.0 | 0.053 |
| MM-FGCN (ours) | 10,674 | **82.6** | 0.054 | 52,546 | **84.4** | 0.058 | 101,046 | **84.5** | 0.61 |

Table 12: Comparison of accuracy and training runtime (in seconds) on Cora datasets. Although our model's runtime is slightly larger than GCN and GAT, our method remains faster than GraphSage and SVD-GCN.

| Model | # Parameters | Acc. ($\uparrow$) | Inference Time ($\downarrow$) | # Parameters | Acc. ($\uparrow$) | Inference Time ($\downarrow$) | # Parameters | Acc. ($\uparrow$) | Inference Time ($\downarrow$) |
|---|---|---|---|---|---|---|---|---|---|
| GCN | 10,246 | 80.4 | **0.011** | 51,238 | 81.2 | **0.015** | 100,204 | 82.0 | **0.017** |
| GAT | 10,124 | 79.6 | 0.019 | 50,422 | 81.7 | 0.020 | 101,684 | 82.5 | 0.023 |
| GraphSage [4] | 10,076 | 78.9 | 0.231 | 51,682 | 80.9 | 0.275 | 107,216 | 81.9 | 0.302 |
| SVD-GCN [15] | 10,578 | 79.6 | 0.227 | 53,412 | 81.4 | 0.247 | 106,452 | 82.5 | 0.258 |
| MM-FGCN (ours) | 10,674 | **82.6** | 0.047 | 52,546 | **84.4** | 0.050 | 101,046 | **84.5** | 0.052 |

Table 13: Comparison of accuracy and inference time (in seconds) under different parameter sizes on the Cora dataset.

## C  RELATED WORKS ON META GRAPH REPRESENTATION LEARNING

Graph representation learning refers to the process of converting the raw graph data into high dimensional vectors while preserving intrinsic graph properties (Chen et al., 2020). Effective graph representations can provide significant insights into graph data and benefit downstream tasks, such as social analysis (Min et al., 2021), molecular property prediction and generation (Zhang et al., 2021; Zang & Wang, 2020), graph generation (Jo et al., 2022; Luo et al., 2023b;a), time series analysis (Li et al., 2021b; Cui et al., 2021), neural signal processing (Ding et al., 2023; Ding & Guan, 2023), etc.

Meta-learning (Hospedales et al., 2022; Liu et al., 2022b; Finn et al., 2017), also referred to as learning to learn, aims to enhance a base learning algorithm through knowledge gained from a meta-algorithm. It has been applied to graph representation learning in (Huang & Zitnik, 2020; Xiao et al., 2021; Zügner & Günnemann, 2019). Huang & Zitnik (2020) use local subgraphs to transfer subgraph-specific information and learn transferable knowledge faster via meta gradients. Xiao et al. (2021) use a meta-learner to relate tasks on graphs describing the relations of their own dimensions to improve few-shot learning. Zügner & Günnemann (2019) optimize the graphs as a hyperparameter using meta-gradients to solve a bi-level optimization problem underlying training-time attacks. While these methods are designed for either transfer learning or multi-task learning scenarios, our proposed meta-learning algorithm for MM-FGCN model aims to improve the base graph representation algorithm, the graph transform of MM-FGCN, by optimizing a meta-algorithm, i.e. the meta-framelet learner. Since a perturbation on graph framelets may cause high variation in graph representations, simultaneously updating the meta-framelet learner and the base model may be inefficient. Therefore our meta-framelet algorithm is learned in a bi-level meta-learning framework to enhance the overall learning performance.

## D  PROOF DETAILS

*Proof of Proposition 2.* According to the discrete tight framelet transform theory (Theorem 2.1 and Theorem 3.1 in (Dong, 2017a)), the series of progressive resolution subspaces $\{V_r\}_r$ with $V_r = \text{span}\left(\{\boldsymbol{\varphi}_{riv}\}_{i,v}\right)$ inherently satisfies the denseness, translation, and dilation properties, making

it a set of desirable multiresolution bases for graph domain data. To complete the proof, one only need to verify the tightness of the MMFS, i.e. $\mathbf{\Phi}_{\text{MM}}\mathbf{\Phi}_{\text{MM}}^{\top}\mathbf{x} = \mathbf{x}$ holds for any graph signal $\mathbf{x}$. Let $\mathcal{I} = ([R] \times [I]) \backslash \{(r, 1) : 1 \leqslant r < R\}$, we have

$$\mathbf{\Phi}_{\text{MM}}\mathbf{\Phi}_{\text{MM}}^{\top} = \sum_{r,i,v} \boldsymbol{\varphi}_{riv}\boldsymbol{\varphi}_{riv} \tag{10}$$

$$= \sum_{(r,i)\in\mathcal{I}} \left(\mathbf{U}g_{r,i}^{\boldsymbol{\omega}}(\gamma^{-J+r}\mathbf{L})\mathbf{U}^{\top}\right)\left(\mathbf{U}g_{r,i}^{\boldsymbol{\omega}}(\gamma^{-J+r}\mathbf{L})\mathbf{U}^{\top}\right)^{\top} \tag{11}$$

$$= \mathbf{U}\left(\sum_{(r,i)\in\mathcal{I}} g_{r,i}^{\boldsymbol{\omega}}{}^{2}(\gamma^{-J+r}\mathbf{L})\right)\mathbf{U}^{\top} \tag{12}$$

$$= \mathbf{U}\left(\left(\sum_{1<i\leqslant I,1\leqslant r<R} g_{1,i}^{\boldsymbol{\omega}}{}^{2}(\gamma^{-J+1}\mathbf{L})\right)\right. \tag{13}$$

$$\left.+ \left(\sum_{i\in[I]} g_{R,i}^{\boldsymbol{\omega}}{}^{2}(\gamma^{-J+R}\mathbf{L})g_{1,1}^{\boldsymbol{\omega}}{}^{2}(\gamma^{-J+R-1}\mathbf{L})\cdots g_{1,1}^{\boldsymbol{\omega}}{}^{2}(\gamma^{-J+1}\mathbf{L})\right)\right)\mathbf{U}^{\top}$$

$$= \mathbf{U}\left(\left(\sum_{1<i\leqslant I,1\leqslant r<R} g_{1,i}^{\boldsymbol{\omega}}{}^{2}(\gamma^{-J+1}\mathbf{L})\right)\right. \tag{14}$$

$$\left.+ \left(g_{1,1}^{\boldsymbol{\omega}}{}^{2}(\gamma^{-J+R-1}\mathbf{L})\cdots g_{1,1}^{\boldsymbol{\omega}}{}^{2}(\gamma^{-J+1}\mathbf{L})\right)\right)\mathbf{U}^{\top}$$

$$= \mathbf{U}\left(\left(\sum_{1<i\leqslant I,1\leqslant r<R} g_{1,i}^{\boldsymbol{\omega}}{}^{2}(\gamma^{-J+1}\mathbf{L})\right) + g_{R-1,1}^{\boldsymbol{\omega}}(\gamma^{-J+R-1}\mathbf{L})\right)\mathbf{U}^{\top} \tag{15}$$

$$= \mathbf{U}\left(\left(\sum_{1<i\leqslant I,1\leqslant r<R-1} g_{1,i}^{\boldsymbol{\omega}}{}^{2}(\gamma^{-J+1}\mathbf{L})\right) + g_{R-2,1}^{\boldsymbol{\omega}}(\gamma^{-J+R-2}\mathbf{L})\right)\mathbf{U}^{\top} \tag{16}$$

$$\vdots \tag{17}$$

$$= \mathbf{U}\left(\sum_{1\leqslant i\leqslant I} g_{1,i}^{\boldsymbol{\omega}}{}^{2}(\gamma^{-J+1}\mathbf{L})\right)\mathbf{U}^{\top} = \mathbf{I}, \tag{18}$$

which completes the proof. ∎

