# OpenReview forum: "Learning Adaptive Multiresolution Transforms via Meta-Framelet-based Graph Convolutional Network"
_ICLR.cc/2024/Conference — ICLR 2024 poster_

### Official Review · Reviewer_PtSQ · 2023-10-30

**Soundness:** 3 good
**Presentation:** 3 good
**Contribution:** 3 good
**Rating:** 6
**Confidence:** 3

**Summary:**

The paper proposes a new method for graph representation learning leverage spectral graph representation learning and meta learning. Experiments show the proposed method shows superior performance on a variety of graph datasets.

**Strengths:**

1. The method is relatively novel, combining several ideas from graph spectral filtering and meta learning.
2. Adequate ablation studies are performed to show the importance of proposed components.

**Weaknesses:**

1. The datasets used in the paper are relatively small, given the fact that there are a numerous large-scale graph benchmark datasets nowadays. It may not be a large concern in the early days, e.g. 2016-2017 when GNN was originally proposed, but in today’s standard more larger graphs are expected to evaluate the proposed approach rigorously and reliably.
2. Lack of some well-known baseline methods such as GIN (Xu, Keyulu, et al. "How powerful are graph neural networks?." arXiv preprint arXiv:1810.00826 (2018).) and graph transformers (e.g. Rampášek, Ladislav, et al. "Recipe for a general, powerful, scalable graph transformer." Advances in Neural Information Processing Systems 35 (2022): 14501-14515.) Given the abundance of such existing methods, I encourage the authors to admit this fact and discuss about the pros and cons of using the proposed MM-FGCN approach in practice.
3. There is no mentioning on the releasement of the code making reproducibility hard.
4. Typo, e.g. “denseness, dilation property, and property (Mallat, 2006)” in Section 3.

**Questions:**

1. Why Geom-GCN in Table 6 lacks std?

---

> ### Author Response · Authors · 2023-11-21
> **Response to Concern 1.**
>
> Thank you for taking the time to review our paper. We sincerely appreciate your thoughtful and constructive feedback on our work. We present our responses as follows.
>
>
> **Concern 1. Extra experiments on large-scale datasets.**
>
> *Weakness 1*: The datasets used in the paper are relatively small, given the fact that there are numerous large-scale graph benchmark datasets nowadays. It may not be a large concern in the early days, e.g. 2016-2017 when GNN was originally proposed, but in today’s standard more larger graphs are expected to evaluate the proposed approach rigorously and reliably.
>
>
>
> **Response to Concern 1. We have conducted extra experiments on large-scale graph benchmark datasets.**
>
> A well-known set of large-scale graph benchmark datasets is the OGBN dataset. Our proposed MM-FGCN can be well applied to the OGBN datasets such as OGBN-arxiv and OGBN-products, and can achieve high performance.
>
> We conduct experiments on OGBN-Arxiv and OGBN-products. To train our MM-FGCN, for OGBN-arxiv, we use the 128-dimensional feature vector as the node feature of each node; for OGBN-product, we use the 100-dimensional feature vector as each node's feature. It is worth noting that some state-of-the-art methods (such as SimTeG+TAPE+RevGAT, TAPE+RevGAT, and SimTeG+TAPE+GraphSAGE) adopt the raw title and abstract form the OGBN-arxiv and detailed product descriptions from OGBN-product as the node features, and achieve better results due to using a good NLP module in extracting node features. Since we mainly compare the performance of the GNN module, we neglect the gain from using NLP modules such as GIANT-XRT or TAPE, etc. Instead, we directly use the preprocessed feature vectors provided in both datasets on our model and our baseline models.
>
> **Training details**
>
> To train the OGBN-Arxiv, we use the full-batch training. In the OGBN-products dataset, we use neighbour-sampling with size = 10 (similar to UniMP [10]) to sample the subgraph during training. Other hyperparameter settings follow our experiments on other datasets as elaborated in the Appendix. We show the corresponding results in the following tables.
>
>
> Corresponding results are shown in **Table 1** and **Table 2**:
>
>
>
>
> | Method     | Test Acc.          | Val. Acc.          |
> |------------|--------------------|--------------------|
> | MLP        | 55.50 $\pm$ 0.23   | 57.65 $\pm$ 0.12   |
> | NODE2VEC [1]   | 70.07 $\pm$ 0.13   | 71.29 $\pm$ 0.13   |
> | GRAPHZOOM [2]  | 71.18 $\pm$ 0.18   | 72.20 $\pm$ 0.07   |
> | P&L + C&S [3]  | 71.26 $\pm$ 0.01   | 73.00 $\pm$ 0.01   |
> | GRAPHSAGE [4]  | 71.49 $\pm$ 0.27   | 72.77 $\pm$ 0.17   |
> | GCN [5]        | 71.74 $\pm$ 0.29   | 73.00 $\pm$ 0.17   |
> | DEEPERGCN [6]  | 71.92 $\pm$ 0.17   | 72.62 $\pm$ 0.14   |
> | SIGN [7]       | 71.95 $\pm$ 0.11   | 73.23 $\pm$ 0.06   |
> | GAAN [8]       | 71.97 $\pm$ 0.18   | –                  |
> | UFGCONV-R [9]  | 71.97 $\pm$ 0.12   | 73.21 $\pm$ 0.05   |
> | UniMP [10]      | 73.11 $\pm$ 0.21| 74.50 $\pm$ 0.05|
> | Exphormer [11]  | 72.44 $\pm$ 0.28   |  –                  |
> | DRGAT [12]      | 74.16 $\pm$ 0.07| 75.34 $\pm$ 0.02|
> | Ours       | **74.25 $\pm$ 0.15**| **75.46 $\pm$ 0.08**|
>
> **Table 1: Experiment on the OGBN-Arxiv dataset.**
>
>
>
>
> | Method               | Test Acc.           | Val. Acc.           |
> |----------------------|---------------------|---------------------|
> | MLP                  | 55.50 $\pm$ 0.23    | 57.65 $\pm$ 0.12    |
> | GCN-Cluster          | 78.97 $\pm$ 0.36    | 92.12 $\pm$ 0.09    |
> | GAT-Cluster          | 79.23 $\pm$ 0.78    | 89.85 $\pm$ 0.22    |
> | GAT-NeighborSampling | 79.45 $\pm$ 0.59    | -                   |
> | GraphSAINT [13]          | 80.27 $\pm$ 0.26    | -                   |
> | DeeperGCN  [6]          | 80.90 $\pm$ 0.20    | 92.38 $\pm$ 0.09    |
> | UniMP [10]               | 82.56 $\pm$ 0.31    | 93.08 $\pm$ 0.17    |
> | AGDN  [14]               | 83.34 $\pm$ 0.27    | 92.29 $\pm$ 0.10    |
> | Ours                 | **84.03 $\pm$ 0.23**| **93.57 $\pm$ 0.12**|
>
> **Table 2: Experiment on the OGBN-Products dataset.**
>
> The experiment results show that our method achieves the highest performance among various state-of-the-art baselines on both datasets.
>
> For the OGBN-Arxiv dataset in which we use full-batch for training, training one epoch takes 2.86 seconds with meta-learning on average, and the inference takes 2.65 seconds. Such a high speed demonstrates the capability of our model in training on large graph datasets.
>
>
> *The aforementioned experiments are added to the Appendix of the manuscript.*

---

> ### Author Response · Authors · 2023-11-21
> **Response to Concern 2. (Part 1/2)**
>
> **Concern 2. Add baselines, and discuss pros and cons.**
>
>  *Weakness 1*: Lack of some well-known baseline methods such as GIN (Xu, Keyulu, et al. "How powerful are graph neural networks?." arXiv preprint arXiv:1810.00826 (2018).) and graph transformers (e.g. Rampášek, Ladislav, et al. "Recipe for a general, powerful, scalable graph transformer." Advances in Neural Information Processing Systems 35 (2022): 14501-14515.) Given the abundance of such existing methods, I encourage the authors to admit this fact and discuss about the pros and cons of using the proposed MM-FGCN approach in practice.
>
>
> **Response to Concern 2.**
>
>
> **Comparison with GIN [16] and graph transformer [17]**
>
> We have added the well-known baseline methods including GIN [16] and Graph transformer [17] in our experiment results. As shown in Table 3, our performance remains to be the SOTA method. These results have been updated to the **Table 1** of the manuscript.
>
>
>
> |              | Cora | Citeseer | PubMed | Cornell | Texas | Wisconsin | Chameleon | Squirrel |
> |--------------|------|----------|--------|---------|-------|-----------|-----------|----------|
> | GIN+0  [16]      | 81.7 $\pm$ 1.3 | 71.4 $\pm$ 0.8 | 79.2 $\pm$ 0.3 |      57.4 $\pm$ 7.8      |  61.4 $\pm$ 5.9     |   58.3 $\pm$ 7.2        |   62.7 $\pm$ 2.7        |  38.0 $\pm$ 1.8      |
> | GIN+ε   [16]     | 81.6 $\pm$ 1.4 | 71.5 $\pm$ 0.8 | 79.1 $\pm$ 0.4 |     59.2 $\pm$ 6.5      |  60.5 $\pm$ 6.2     |   61.1 $\pm$ 6.8        |   61.4 $\pm$ 2.2  |  37.2 $\pm$ 1.5    |
> GraphGPS  [17]   |   83.1  $\pm$ 0.7  |    72.3 $\pm$ 0.8      |  79.5 $\pm$ 0.4 | 67.2 $\pm$ 7.7 | 79.5 $\pm$ 5.6 | 76.9 $\pm$ 4.9 | 68.2 $\pm$ 2.5 | 58.4 $\pm$ 1.6    |
> | MM-FGCN (Ours)    |  **84.4 $\pm$ 0.5** | **73.9 $\pm$ 0.6** |  **80.7 $\pm$ 0.2** |  **88.9 $\pm$ 8.3** |  **86.1 $\pm$ 4.5** |  **88.5 $\pm$ 4.1** |  **73.9 $\pm$ 2.1** |  **67.5 $\pm$ 1.2** |
>
> **Table 3: Comparison with GIN [16] and GraphGPS [17] on the node classification task.**
>
>
>
> **Pros and Cons of MM-FGCN.**
>
> The pros of our method can be summarized as follows:
>
> - Adaptive: The meta-multiresolution framelet transform of our method is adaptively tailored to diverse graph instances and distributions, which enhances our model's performance on various types of graphs, such as assortative graphs and disassortative graphs.
>
> - Enables multi-resolution representations: Through analysing the graph features at multiple resolutions, our model provides a comprehensive representation of the graph data and can achieve significantly high performance in downstream tasks, such as node and graph classification.
>
>
>
> The Cons of our method can also be summarized as follows:
>
> - Meta-training is needed: To achieve the best performance of our method, the meta-learning is required. However, the meta-learning requires additional computational resources and can moderately slow down the training speed of our model.
>
> - Extra parameters are needed: Compared to several well-known graph models such as GCN and GAT, our model introduces more trainable parameters to learn the meta-multiresolution framelets, requiring extra computational overhead. In the following section, we evaluate the training and inference time cost of MM-FGCN, showing that our model achieves SOTA performance without significant overhead.
>
> - Chebyshev approximation error exists: We leverage Chebyshev approximation to accelerate the computation of the meta-framelet transform. In practice, we show that the 4-ordered Chebyshev approximation is adequate for achieving saturated high performance. However, in theory, the Chebyshev approximation error exists, leading to a non-zero violation of the tightness constraint.

---

> > ### Author Response · Authors · 2023-11-21
> > **Response to Concern 3 and Concern 4**
> >
> > **Concern 3. Code release statement.**
> >
> > *Weaknesses 3:* There is no mentioning on the releasement of the code making reproducibility hard.
> >
> > **Response to Concern 3.**
> >
> > We will release the code upon the acceptance of this paper.
> >
> >
> > **Concern 4. Typo, and Geom-GCN lack of std.**
> >
> >
> > **Response to Concern 4.**
> >
> > We have revised the typo in the manuscript, and we have added the std of Geom-GCN in the revised paper.
> >
> >
> >
> > ----------------------------------------------------------------------------------------------------------------------------------------------------
> >
> >
> > **References**
> >
> > [1] Grover, Aditya, and Jure Leskovec. "node2vec: Scalable feature learning for networks." Proceedings of the 22nd ACM SIGKDD international conference on Knowledge discovery and data mining. 2016.
> >
> > [2] Deng, Chenhui, et al. "Graphzoom: A multi-level spectral approach for accurate and scalable graph embedding." arXiv preprint arXiv:1910.02370 (2019).
> >
> > [3] Huang, Qian, et al. "Combining label propagation and simple models out-performs graph neural networks." ICLR 2021.
> >
> > [4] Hamilton, Will, Zhitao Ying, and Jure Leskovec. "Inductive representation learning on large graphs." Advances in neural information processing systems 30 (2017).
> >
> > [5] Kipf, Thomas N., and Max Welling. "Semi-supervised classification with graph convolutional networks." arXiv preprint arXiv:1609.02907 (2016).
> >
> > [6] Li, Guohao, et al. "Deepergcn: All you need to train deeper gcns." arXiv preprint arXiv:2006.07739 (2020).
> >
> > [7] Rossi, Emanuele, et al. "Sign: Scalable inception graph neural networks." arXiv preprint arXiv:2004.11198 7 (2020)
> >
> > [8] Zhang, Jiani, et al. "Gaan: Gated attention networks for learning on large and spatiotemporal graphs." arXiv preprint arXiv:1803.07294 (2018).
> >
> > [9] Zheng, Xuebin, et al. "How framelets enhance graph neural networks." arXiv preprint arXiv:2102.06986 (2021).
> >
> > [10] Shi, Yunsheng, et al. "Masked label prediction: Unified message passing model for semi-supervised classification." arXiv preprint arXiv:2009.03509 (2020).
> >
> > [11]  Shirzad, Hamed, et al. "Exphormer: Sparse transformers for graphs." arXiv preprint arXiv:2303.06147 (2023).
> >
> > [12] Zhang, Lei, et al. "DRGCN: Dynamic Evolving Initial Residual for Deep Graph Convolutional Networks." arXiv preprint arXiv:2302.05083 (2023).
> >
> > [13] Zeng, Hanqing, et al. "Graphsaint: Graph sampling based inductive learning method." arXiv preprint arXiv:1907.04931 (2019).
> >
> > [14] Sun, Chuxiong, et al. "Adaptive Graph Diffusion Networks." arXiv preprint arXiv:2012.15024 (2020).
> >
> > [15] Entezari, Negin, et al. "All you need is low (rank) defending against adversarial attacks on graphs." Proceedings of the 13th International Conference on Web Search and Data Mining. 2020.
> >
> > [16] Xu, Keyulu, et al. "How powerful are graph neural networks?." arXiv preprint arXiv:1810.00826 (2018).
> >
> > [17] Rampášek, Ladislav, et al. "Recipe for a general, powerful, scalable graph transformer." Advances in Neural Information Processing Systems 35 (2022): 14501-14515.

---

> ### Author Response · Authors · 2023-11-21
> **Response to Concern 2. (Part 2/2)**
>
> **Training and inference time cost**
>
> One of the main limitations of our method is that due to adding new additional components, the runtime complexity may be increased. To investigate the training speed in practice, we compare the training and inference cost of our MM-FGCN against training other baselines, including GCN, GAT, GraphSage and SVD-GCN. We report the time cost of one training epoch in Table 3, and we evaluate the time cost of on single forward pass in Table 4. Specifically, each training epoch of MM-FGCN includes both the base- and the meta-update step. And we set the Chebyshev approximation order $k$ and the number of meta-framelets $r$ as $k=4$ and $r=4$. We conduct the experiments on varying GNN architectures, with varying parameter budgets (e.g. model size). The experiments are conducted on a single 40G A100 GPU.
>
> As shown in Table 3 and Table 4, given the same parameter budget, our MM-FGCN achieves significantly higher performance than GCN and GAT within a moderate computational overhead. When compared to SVD-GCN and GraphSage, our model achieves higher performance in both training and inference speed and accuracy.
>
> | Model | \# parameters | Acc. ($\uparrow$) | Time/Epoch ($\downarrow$) | \# parameters | Acc. ($\uparrow$) | Time/Epoch ($\downarrow$) | \# parameters | Acc. ($\uparrow$) | Time/Epoch ($\downarrow$) |
> |-------------------|-----------|---------|-----------------------------|-----------|---------|-----------------------------|----------|----------|-----------------------------|
> | GCN          |   10,246  | 80.4  |**0.012**  | 51,238       | 81.2          | **0.016**               |  100,204  | 82.0               | **0.018**               |
> | GAT           | 10,124   | 79.6             | 0.021       | 50,422                 | 81.7               | 0.022             | 101,684          | 82.5               | 0.024                       |
> | GraphSage [4]    | 10,076    | 78.9               | 0.248                | 51,682        | 80.9               | 0.281             | 107,216          | 81.9               | 0.315                       |
> | SVD-GCN [15]      | 10,578   | 79.6               | 0.236  |           53,412          | 81.4               | 0.254             | 106,452          | 82.5               | 0.261                     |
> | MM-FGCN (ours) w/o meta-training | 10,674  |  82.2 | 0.048        | 52,546               | 83.7               | 0.050               | 101,046        | 84.0               | 0.053                       |
> | MM-FGCN (ours)   |  10,674 | **82.6**       | 0.054          | 52,546             | **84.4**       | 0.058         | 101,046              | **84.5**       | 0.61                       |
>
> **Table 3**: Comparison of accuracy and **training** runtime (in seconds) on Cora datasets. Although our model's runtime is slightly larger than GCN and GAT, our method remains faster than GraphSage and SVD-GCN.
>
>
>
>
> | Model | \# parameters | Acc. ($\uparrow$) | Inference time ($\downarrow$) | \# parameters | Acc. ($\uparrow$) | Inference time ($\downarrow$) | \# parameters | Acc. ($\uparrow$) | Inference time ($\downarrow$) |
> |-------------------|----------|----------|--------------------------------|------------|--------|--------------------------------|---------|-----------|--------------------------------|
> | GCN          | 10,246     | 80.4               | **0.011**             | 51,238     | 81.2               | **0.015**       | 100,204           | 82.0               | **0.017**                  |
> | GAT        | 10,124       | 79.6               | 0.019              | 50,422            | 81.7               | 0.020              | 101,684            | 82.5               | 0.023                          |
> | GraphSage [4]    | 10,076    | 78.9               | 0.231                 | 51,682         | 80.9               | 0.275                 | 107,216         | 81.9               | 0.302                          |
> | SVD-GCN [15]     | 10,578     | 79.6               | 0.227                |53,412          | 81.4               | 0.247                | 106,452          | 82.5               | 0.258                          |
> | MM-FGCN (ours)  | 10,674  | **82.6**       | 0.047             | 52,546             | **84.4**       | 0.050             | 101,046             | **84.5**       | 0.052                          |
>
> **Table 4**: Comparison of accuracy and **inference** time (in seconds) under different parameter sizes on the Cora dataset.
>
>
> To summarize, our can achieve high performance on various types of graphs. Although, comparing the training and inference speed, our model is slower than GCN and GAT, the speed of our method is relatively fast compared to some other baselime methods.

---

### Official Review · Reviewer_sJoC · 2023-10-30

**Soundness:** 3 good
**Presentation:** 3 good
**Contribution:** 3 good
**Rating:** 8
**Confidence:** 3

**Summary:**

This paper proposes a Multiresolution Meta-Framelet-based Graph Convolutional Networks (MM-FGCN), which employs a diverse set of framelets for constructing graph convolution and learns meta-framelet generator networks via the meta-learning scheme. Since most GNNs depend on single-resolution graph feature extraction, they often fail to capture local patterns and community patterns simultaneously. To resolve it, some papers have proposed multi-resolution graph feature extraction-based GNNs. But, they use predefined and hand-crafted multiresolution transforms. To address these issues, this paper designs meta-learning based adaptive multiresolution graph convolution and they have shown the effectiveness with their experiments.

**Strengths:**

- The proposed paper deals with really important research problem. Simultaneously capturing high and low resolution with graph convolution is interesting and important.
- The proposed method seems novel to me.
- From their experiments, the proposed meta-framelet-based graph convolutional networks show good performance on various tasks.
- The paper is well-written and easy to follow. In particular, the preliminary section provides the necessary and detailed information to understand the proposed method.

**Weaknesses:**

- One of the important details about how to split the meta dataset from the main dataset is missing. It should have been provided to fairly compare the proposed method with other graph neural networks.
- In the same context, it would be better if you explain why meta learning framelet transforms is better than directly training them. From Table 3, it is easy to know that framelet transforms with meta-learning is more effective compared to the direct training scheme. But, why meta-learning scheme is better than the direct training scheme is mysterious. I think this is related to how to construct the meta dataset.
- It would be better if the author provided the change of filters according to the training step since the author claimed that the limitation of existing multiresolution transforms works is that they remain fixed throughout the training process. So, I'd like to see the variations of filters during the training and the analysis about it.

**Questions:**

- I think MM-FGCN can also be applied to the graph classification tasks. Could you provide the performance of MM-FGCN with MM-FGPool and MMFGCN with the standard graph pooling?

---

> ### Author Response · Authors · 2023-11-22
> **Response to Concern 1.**
>
> Thank you for taking the time to review our paper. We sincerely appreciate your thoughtful and constructive feedback on our work. We present our responses as follows.
>
>
> **Concern 1. Performance of MMFGCN with the standard graph pooling.**
>
> *Question 1*: I think MM-FGCN can also be applied to the graph classification tasks. Could you provide the performance of MM-FGCN with MM-FGPool and MMFGCN with the standard graph pooling?
>
>
> **Response to Concern 1. MM-FGPool outperforms the standard graph pooling.**
>
> Yes, MM-FGCN can also be applied to the graph classification task with standard graph pooling. However, the performance is slightly lower than the MM-FGPool. We show the performance of MM-FGCN with the standard graph pooling compared with MM-FGPool in the following Table.
>
> |                     | PROTEINS (↑) | Mutagenicity (↑) | D&D (↑) | NCI1 (↑) | Ogbg-molhiv (↑) | QM7 (↓) |
> |---------------------|--------------|-------------------|---------|----------|-----------------|---------|
> | Standard graph pooling|      77.79    $\pm$ 2.47    |        83.06 $\pm$ 1.35           |  81.04 $\pm$ 1.61       |  78.14 $\pm$ 0.93       |   78.62 $\pm$ 0.87             |    41.56 $\pm$ 0.91    |
> | MM-FGPool           | **78.07 $\pm$ 2.36** | **83.91 $\pm$ 1.32** | **81.51 $\pm$ 1.55** | **78.57$\pm$ 0.82** | **79.12 $\pm$ 0.85** | **41.19 $\pm$ 0.88** |
>
> **Table 1: Comparison of MM-FGCN with MM-FGPool and standard graph pooling**

---

> ### Author Response · Authors · 2023-11-22
> **Response to Concern 2.**
>
> **Concern 2**. How to split the meta dataset?
>
> *Weakness 1*: One of the important details about how to split the meta dataset from the main dataset is missing. It should have been provided to fairly compare the proposed method with other graph neural networks.
>
>
> **Response to Concern 2.**
>
> The $S_{meta}$ and $S_{main}$ are the data randomly split from the **training data**. In our experiment, we take 80% of the training data as the $S_{main}$ and 20% as $S_{meta}$.
>
> In practice, our MM-FGCN shares the same train and test dataset splitting scheme as other baseline methods. We first train the MM-FGCN by meta-training (Algorithm 2) on the training data. Then, we fix the trained parameters $\theta, \xi$, and we evaluate the accuracy of the MM-FGCN on the testing dataset. The dataset splitting in line 3 of Algorithm 2 is done within the training data. Hence, in our experiments, the MM-FGCN is evaluated under the same data resource as other baselines.

---

> ### Author Response · Authors · 2023-11-22
> **Response to Concern 3.**
>
> **Concern 3**. Why the meta-learning framelet transform is better than directly training them?
>
> *Weakness 2*: In the same context, it would be better if you explain why meta learning framelet transforms is better than directly training them. From Table 3, it is easy to know that framelet transforms with meta-learning is more effective compared to the direct training scheme. But, why meta-learning scheme is better than the direct training scheme is mysterious. I think this is related to how to construct the meta dataset.
>
> **Response to Concern 3.**
>
>
> In a nutshell, using meta-learning improves the performance of MM-FGCN on various graph learning scenarios. As shown in **Table 2** and **Table 3**, our ablation study proves that using meta-learning paradigm leads to higher performance compared to directly updating all the parameters $\theta, \xi$. In intuition, meta-learning encourages MM-FGCN to learn meta-framelet representations that generalize well.
>
>
>
> **Ablation studies on meta-learning**
>
> We compare the performance of MM-FGCN, both with and without meta-training, on the node classification datasets including Cora, Citeseer, PubMed, Cornell, Texas, Wisconsin, Chameleon, and Squirrel; and graph classification datasets including PROTEINS, Mutagenicity, D&D, NCI1, Ogbg-molhiv, and QM7. The experiments are conducted under the same experiment settings described in Appendix A2. It is important to note that the main- and meta-dataset was **split randomly** at a ratio of 80%:20%, without any deliberate design in the splitting scheme.
>
>
> We conduct extra experiments to show the performance of with and without meta-learning in the following tables:
>
> |                     | Cora | Citeseer | PubMed | Cornell | Texas | Wisconsin | Chameleon | Squirrel |
> |---------------------|------|----------|--------|---------|-------|-----------|-----------|----------|
> | MM-FGCN w/o meta-learning|    82.9 ± 0.5 | 72.2 ± 0.7 | 80.1 $\pm$ 0.2 |    78.2 ± 8.5 | 77.9 ± 9.2 |  82.3 $\pm$ 4.8  |62.6 ± 2.6 | 59.2 ± 2.2     |
> | MM-FGCN           |  **84.4 $\pm$ 0.5** | **73.9 $\pm$ 0.6** | **80.7 $\pm$ 0.2** | **88.9 $\pm$ 8.3** | **86.1 $\pm$ 4.5** | **88.5 $\pm$ 4.1** | **73.97 $\pm$ 2.1** | **67.5 $\pm$ 1.2** |
>
> **Table 2: Comparison of with and without meta-learning framework on the node classification tasks.**
>
>
>
>
> |      | PROTEINS (↑) | Mutagenicity (↑) | D&D (↑) | NCI1 (↑) | Ogbg-molhiv (↑) | QM7 (↓) |
> |-----------------------------|--------------|-------------------|---------|----------|-----------------|---------|
> | MM-FGPool w/o meta-learning |   77.86 $\pm$ 2.53 | 82.3 ± 1.4        |    81.0 ± 1.7 | 75.9 ± 0.9                 |   78.57 $\pm$ 0.73       |   41.65 $\pm$  0.91             |
> | MM-FGPool                   | **78.07 $\pm$ 2.36** | **83.91 $\pm$ 1.32** | **81.51 $\pm$ 1.55** | **78.57 $\pm$ 0.82** | **79.12 $\pm$ 0.85** | **41.19 $\pm$ 0.88** |
>
> **Table 3: Comparison of with and without meta-learning framework on the graph classification tasks.**
>
>
>
> As shown in Table [1] and Table [2], in practice, using meta-training (Algorithm 2) leads to a consistent performance gain on varying datasets in both graph and node classification settings. Moreover, as shown in Figure [8] of the paper, the meta-training process accelerates the convergence of MM-FGCN and improves the generalization performance of MM-FGCN.
>
>
> **Discussions on meta-learning**
>
> In intuition, using different data batches in the meta-step and main-step of Algorithm 2 helps MM-FGCN to avoid overfitting. In the meta-step, the meta-framelet learner $\xi$ is first optimized to improve the meta-framelet for the current backbone model $\theta$. Based on this optimized meta-framelet representation, the whole model is then evaluated and optimized with another newly sampled data batch. Using two identically distributed but distinct data batches, the $\xi$-meta-framelet transform will be cautiously updated until achieving good performance on both the meta- and main-data batches. Consequently, we anticipate that the meta-training process encourages MM-FGCN to find meta-framelet representations that can be generalized to different batches drawn from the underlying distribution, rather than merely following the gradient calculated on the current data batch, thus reducing the risk of overfitting.

---

> ### Author Response · Authors · 2023-11-22
> **Response to Concern 4.**
>
> **Concern 4**. Change of filters according to the training step.
>
> *Weakness 3*: It would be better if the author provided the change of filters according to the training step since the author claimed that the limitation of existing multiresolution transforms works is that they remain fixed throughout the training process. So, I'd like to see the variations of filters during the training and the analysis about it.
>
>
> **Response to Concern 4.**
>
>
> In the figure of the Appendix: "Variation of filters during the training process", we visualize the variation of filters during the training process on the Cora and Cornell datasets.
> We train our model on both datasets for 100 epochs and visualize the filters at training steps 0, 20, 50, and 100.
>
>
>
>
> **Empirical analysis on Cora**
>
>
>
> Recall that the Cora dataset contains a graph of a citation network, where each node represents a scientific publication and the node labels are the associated research domain. Thus, the node label (i.e. the graph signal of interest) varies smoothly across the nodes, emphasizing the importance of low-frequency information in learning good graph representations.
>
> As shown in the figure of the Appendix: "Variation of filters during the training process", we observe that MM-FGCN automatically learns to focus more on establishing the low-frequency representations: at the initial stage of the training process, the meta-filters are evenly separated and do not show special preference to the graph signals at specific frequencies.  During the training process, the filters tend to concentrate on the low-frequency part of the graph signal to retrieve more refined low-frequency information of the graph.
>
>
> **Empirical analysis on Cornell**
>
> The Cornell dataset contains a website network, where each node represents a web page that is manually classified into five categories: student, project, course, staff, and faculty. Edges represent hyperlinks between them. In this case, nodes with the same labels are not likely to link to each other. Thus, the node label (i.e. the graph signal of interest) varies drastically across neighbouring nodes, showing that high-frequency information is essential in learning good graph representations.
>
>
> As shown in the figure of the Appendix: "Variation of filters during the training process", we observe that MM-FGCN automatically learns to focus more on establishing the high-frequency representations: at the initial stage of the training process, the meta-filters are evenly separated and do not show special preference to the graph signals at specific frequencies. During the training process, the filters tend to concentrate on the high-frequency part of the graph signal to refine the high-frequency representations of the graph.

---

> > ### Comment · Reviewer_sJoC · 2023-11-22
> > **Thank you for the response**
> >
> > I really appreciate the detailed response to my concerns and a question.
> > My concerns are fully addressed with the response and I have read all the reviews and their corresponding responses.
> >
> > As other reviewers commented, I think the proposed method has a small; practical issue such as the time and space complexity. But, I think that the contribution of this work is enough to get the acceptance on the top-tier machine learning conference. The application of the meta-learning algorithm to the framelet-based graph convolution is really interesting and seems novel.
> > In addition, the paper is easily written to follow and provides detailed explanations about their proposed method.
> >
> > So, I decided to raise my score from 6 to 8.

---

### Official Review · Reviewer_oTNL · 2023-10-31

**Soundness:** 3 good
**Presentation:** 3 good
**Contribution:** 3 good
**Rating:** 8
**Confidence:** 4

**Summary:**

The Multiresolution Meta-Framelet-based Graph Convolutional Network (MM-FGCN) is a novel approach to graph representation learning that allows for adaptive multiresolution analysis across diverse graphs. It achieves state-of-the-art performance on various graph learning tasks.

In my opinion, adaptive multi-resolution is a promising yet difficult approach for the representation of graphs, lying in the intersection of graph signal processing and graph machine learning. In particular, I appreciate the construction of the framelet filters to satisfy the three properties (where denseness is different from the sparse and redundant representation in wavelet and framelet). The paper extends the framelet theory and presents the solution with relatively few parameters (only $\Theta, \omega$) to avoid a large number of parameters for the framelet system, yet achieves significant improvement in the experiments. The theories and proofs are solid in general, and though some of the writing could be improved, I think it is an excellent piece of work.

**Strengths:**

1. The paper proposes a novel approach to design an adaptive learnable set of multi-resolution representations on graphs, with solid theoretical motivation and proof.
2. The proposed method significantly improves node classification (especially on disassortative tasks) and graph classification tasks.

**Weaknesses:**

1.The paper lacks some necessary model  and data descriptions for the implementation, like the specification of neural network $M_\xi$.
2. minor issues:  missing "translation" before property near "Mallat, 2006".

**Questions:**

1. How is the neural network $M_\xi$ formulated? What does the output $\omega$ look like?
2. How is the graph data split to obtain $S_{meta}, S_{main}$?
3. Why is the meta training needed for the training procedure?

---

> ### Author Response · Authors · 2023-11-22
> **Response to Concern 1.**
>
> Thank you for taking the time to review our paper. We sincerely appreciate your thoughtful and constructive feedback on our work. We present our responses as follows.
>
> **Concern 1. Formulation of $M_{\xi}$ and the ouput $\omega$.**
>
> *Weakness 1*: The paper lacks some necessary model and data descriptions for the implementation, like the specification of neural network $M_{\xi}$.
>
> *Question 1*: How is the neural network $M_{\xi}$ formulated? What does the output $\omega$ look like?
>
> **Response to Concern 1.**
>
>
> *The following discussion is added to the Appendix.*
>
> In practice, the neural network $M_{\xi}$ is set to be a 2-layer standard GCN network that maps a graph data $(X,A)\in R^{n\times d} \times R^{n\times n}$ into a $2(I-1)$-dimensional vector $\omega\in R^{2(I-1)}$, i.e. the parameters of the meta-filters $\{g_{r,i}^{\mathbf{\omega}}\}_{r,i}$.
>
> Specifically, we set $M_{\xi}(X,A)=1/n \cdot 1^\top \sigma(L\ \sigma(LXW_1)W_2)W_3$, where $W_1\in R^{d\times d/2}, W_2\in R^{d/2\times d/2}$ and $W_3\in R^{d/2\times 2(I-1)}$ are trainable weights, $L$ is the normalized graph Laplacian matrix computed from the adjacency matrix $A$, and $\sigma(\cdot)$ is the ReLU activation function. We denote the collection all the trainable weights as $\xi=(W_1,W_2, W_3)$.
>
> After obtaining the $2(I-1)$-dimensional output $\omega\in R^{2(I-1)}$ from $M_{\xi}$, we use Equation (3) and Equation (4) to determine the exact formulation of the meta-filters $g^{\omega}_{r,i}$.
>
> To determine the formulation of $(r,i)$-th meta-filter, we first decode its lower- and the upper-cutoff positions and margins, i.e. $c_i$, $c_{i+1}$, $\epsilon_i$, and $\epsilon_{i+1}$, from $\omega$ using Equation (3). Then, the formulation of $g_{1,i}^\omega$ (the bandpass starting from $c_{i}+\epsilon_i$ to $c_{i+1}+\epsilon_{i+1}$) can be directly derived from Equation (4). Finally, starting from $g_{1,i}^{\omega}$, we can determine $g_{r,i}^{\omega}$ in an iterative manner, using the second condition in Proposition 2. The visualization example of the meta-filters is shown in Figure 1 and Figure 2 of the manuscript.

---

> ### Author Response · Authors · 2023-11-22
> **Response to Concern 2.**
>
> **Concern 2. Data splitting for $S_{meta}$ and $S_{main}$**
>
>
> *Question 2*: How is the graph data split to obtain $S_{meta}$ and $S_{main}$?
>
> **Response to Concern 2.**
>
>
> The $S_{meta}$ and $S_{main}$ are the data randomly split from the training data. In our experiment, we take 80% of the training data as the $S_{main}$ and 20% as $S_{meta}$.
>
> In practice, our MM-FGCN shares the same train and test dataset splitting scheme as other baseline methods. We first train the MM-FGCN by meta-training (Algorithm 2) on the training data. Then, we fix the trained parameters $\theta, \xi$, and we evaluate the accuracy of the MM-FGCN on the testing dataset. The dataset splitting in line 3 of Algorithm 2 is done randomly within the training data. Hence, in our experiments, the MM-FGCN is evaluated under the same data resource as other baselines.

---

> ### Author Response · Authors · 2023-11-22
> **Response to Concern 3.**
>
> **Concern 3. The need for meta-training.**
>
> *Question 3*: Why is the meta training needed for the training procedure?
>
> **Response to Concern 3. Meta-learning improves the generalization performance and encourages convergence.**
>
> In a nutshell, using meta-learning improves the performance MM-FGCN on various graph learning scenarios. In intuition, meta-learning encourages MM-FGCN to learn meta-framelet representations that generalize well.
>
> **Ablation studies on meta-learning**
>
> We compare the performance of MM-FGCN, both with and without meta-training, on the node classification datasets including Cora, Citeseer, PubMed, Cornell, Texas, Wisconsin, Chameleon, and Squirrel; and graph classification datasets including PROTEINS, Mutagenicity, D&D, NCI1, Ogbg-molhiv, and QM7. The experiments are conducted under the same experiment settings described in Appendix A2. It is important to note that the main- and meta-dataset was **split randomly** at a ratio of 80%:20%, without any deliberate design in the splitting scheme.
>
>
> We conduct extra experiments to show the performance of with and without meta-learning in the following tables:
>
> |                     | Cora | Citeseer | PubMed | Cornell | Texas | Wisconsin | Chameleon | Squirrel |
> |---------------------|------|----------|--------|---------|-------|-----------|-----------|----------|
> | MM-FGCN w/o meta-learning|    82.9 ± 0.5 | 72.2 ± 0.7 | 80.1 $\pm$ 0.2 |    78.2 ± 8.5 | 77.9 ± 9.2 |  82.3 $\pm$ 4.8  |62.6 ± 2.6 | 59.2 ± 2.2     |
> | MM-FGCN           |  **84.4 $\pm$ 0.5** | **73.9 $\pm$ 0.6** | **80.7 $\pm$ 0.2** | **88.9 $\pm$ 8.3** | **86.1 $\pm$ 4.5** | **88.5 $\pm$ 4.1** | **73.97 $\pm$ 2.1** | **67.5 $\pm$ 1.2** |
>
> **Table 1: Comparison of with and without meta-learning framework on the node classification tasks.**
>
>
>
>
> |      | PROTEINS (↑) | Mutagenicity (↑) | D&D (↑) | NCI1 (↑) | Ogbg-molhiv (↑) | QM7 (↓) |
> |-----------------------------|--------------|-------------------|---------|----------|-----------------|---------|
> | MM-FGPool w/o meta-learning |   77.86 $\pm$ 2.53 | 82.3 ± 1.4        |    81.0 ± 1.7 | 75.9 ± 0.9                 |   78.57 $\pm$ 0.73       |   41.65 $\pm$  0.91             |
> | MM-FGPool                   | **78.07 $\pm$ 2.36** | **83.91 $\pm$ 1.32** | **81.51 $\pm$ 1.55** | **78.57 $\pm$ 0.82** | **79.12 $\pm$ 0.85** | **41.19 $\pm$ 0.88** |
>
> **Table 2: Comparison of with and without meta-learning framework on the graph classification tasks.**
>
>
>
> As shown in Table [1] and Table [2], in practice, using meta-training (Algorithm 2) leads to a consistent performance gain on varying datasets in both graph and node classification settings. Moreover, as shown in Figure [8] of the paper, the meta-training process accelerates the convergence of MM-FGCN and improves the generalization performance of MM-FGCN.
>
>
> **Discussions on meta-learning**
>
> In intuition, using different data batches in the meta-step and main-step of Algorithm 2 helps MM-FGCN to avoid overfitting. In the meta-step, the meta-framelet learner $\xi$ is first optimized to improve the meta-framelet for the current backbone model $\theta$. Based on this optimized meta-framelet representation, the whole model is then evaluated and optimized with another newly sampled data batch. Using two identically distributed but distinct data batches, the $\xi$-meta-framelet transform will be cautiously updated until achieving good performance on both the meta- and main-data batches. Consequently, we anticipate that the meta-training process encourages MM-FGCN to find meta-framelet representations that can be generalized to different batches drawn from the underlying distribution, rather than merely following the gradient calculated on the current data batch, thus reducing the risk of overfitting.

---

> ### Author Response · Authors · 2023-11-22
> **Response to Concern 4.**
>
> **Concern 4.**
>
> minor issues: missing "translation" before property near "Mallat, 2006".
>
>
> **Response to Concern 4.**
>
>
> We have revised the paper and added the translation before "property" near "Mallat, 2006".
>
>
>
> -------------------------------------------------------------------------------------------------------------------------------------------
> **References**
>
>
> [1]  Han, Bin, Zhenpeng Zhao, and Xiaosheng Zhuang. "Directional tensor product complex tight framelets with low redundancy." Applied and Computational Harmonic Analysis 41.2 (2016): 603-637.
>
> [2] Chelsea Finn, Pieter Abbeel, and Sergey Levine. Model-agnostic meta-learning for fast adaptation of deep networks. In International conference on machine learning, pp. 1126–1135. PMLR, 2017.

---

> > ### Comment · Reviewer_oTNL · 2023-11-22
> >
> > Thanks for your great efforts and detailed response. In general, my main concerns are addressed. Though I have not seen the practices of meta-learning on the batches of identical distributions, I think it could make some sense since different graph structures bring different eigenspace on multiple graphs and given the significant improvement of meta-training on the experiments present in the response.
> > I hope the authors could supplement the missing necessary details in the article according to our discussion, such as the neural network $\mathcal{M}_\xi$, data split of meta-learning, etc., to enhance the readability of the article. I would raise my score to 8.

---

> > > ### Author Response · Authors · 2023-11-22
> > > **Response to Official Comment by Reviewer oTNL**
> > >
> > > Thank you so much for reevaluating our paper and increasing the score. Your valuable feedback and support are greatly appreciated.
> > >
> > > We have added the missing necessary details to the manuscript. In **Appendix A.2**, we have added the subsections of "Design of the neural network $M_{\xi}$", and "Split of $S_{meta}$ and $S_{main}$". In **Appendix  B.8 "Ablation studies of meta-learning"**, we have added the extra experiments on the ablation study of meta-learning and discussed why the meta-learning can enhance our model's performance.
> > >
> > > Thanks again for your time and effort in reviewing our paper.

---

### Official Review · Reviewer_nDyN · 2023-10-31

**Soundness:** 3 good
**Presentation:** 3 good
**Contribution:** 2 fair
**Rating:** 6
**Confidence:** 4

**Summary:**

This paper use meta-learning to build for adaptive framelet for GNN.

They evaluate the model's performance on various graph learning tasks.

**Strengths:**

1. MM-FGCN is can learn adaptive multiresolution representation.

2. Achieve STOA performance on graph learning tasks.

3. parameterization of the meta-framelet generator uses use fewer parameters than previous method.

**Weaknesses:**

1. Require additional meta training.
2. The framelet is defined using a meta-band-pass filter based on polynomial splines, the model use Chebyshev approximation to circumventing the need for eigen-decomposition, but compared with graph-structure-based approach, Chebyshev approximation seems a more expensive.
3. Following the previous point, it seems it's hard to use the model on large graphs. (ogbn tasks)
4. The performance is not close to STOA, recent studies show better performance, e.g. [1],[2]

[1] https://arxiv.org/abs/2305.10498
[2] https://arxiv.org/abs/2105.07634

**Questions:**

1. Could you provide details on the time cost for meta-learning as well as the inference time for the model?
2. Would it be possible to evaluate the model on larger datasets like ogbn? I did observe ogbg-molhiv, but I'm referring to graphs with a larger number of nodes rather than the total number of graphs.
3. An detailed introduction to meta-learning would help readers in gaining a clearer understanding of the experimental setup.

---

> ### Author Response · Authors · 2023-11-20
> **Response to Concern 1. (Part 1/2)**
>
> Thank you for taking the time to review our paper. We sincerely appreciate your thoughtful and constructive feedback on our work. We present our responses as follows.
>
>
>
> **Concern 1. The time cost of training and inference with MM-FGCN.**
>
> *Question 1.* Could you provide details on the time cost for meta-learning as well as the inference time for the model?
>
> *Weakness 2.* The framelet is defined using a meta-band-pass filter based on polynomial splines, the model use Chebyshev approximation to circumventing the need for eigen-decomposition, but compared with graph-structure-based approach, Chebyshev approximation seems a more expensive.
>
> **Response to Concern 1. The time cost of training and inference with MM-FGCN is comparable to other baselines.**
>
> Using meta-learning and Chebyshev approximation does **NOT** significantly increase the training and inference time; rather, it can boost our model's performance on various graph learning tasks. We provide both theoretical and empirical justifications.
>
> **Time complexity analysis**
>
> The complexity of a single forward pass of the meta-framelet convolution is approximately $r$ times as the cost of standard graph convolution, where $r$ is the number of meta-framelets. In practice, our MM-FGCN can be computed efficiently, without significant overhead.
>
> Suppose we instantiate the meta-learner $\mathcal{M}_{\xi}$ as a $L$-layer GCN, we use a $k$-order Chebyshev approximation for the $n\times n$ graph Laplacian matrix $\mathbf{L}$, and we set the number of filters as $r$. Then, for each $d$-dimensional input graph feature $\mathbf{H}\in \mathbb{R}^{n\times d}$, the computational complexity of meta-framelet convolution is $O(rn\times n \times d) + O(n\times n\times d) + O(r\times k^2 \times n^2)=O(rn^2(d+k^2))$. In this setting, the standard graph convolutional layer costs $O(n^2d)$.
>
> As shown in Table 1 and Table 2 below, our MM-FGCN is able to achieve the SOTA performance with $k=4$ and $r=4$, without significant computational overhead.
>
> **Training and inference time cost**
>
> In the following **Table 1** and **Table 2** **(Response to Concern 1. (Part 2/2))**, we compare both the training and inference time cost of our MM-FGCN against training other baselines, including GCN, GAT, GraphSage and SVD-GCN [15]. We report the time cost of one training epoch in Table 1, and we evaluate the time cost of on single forward pass in Table 2. Specifically, each training epoch of MM-FGCN includes both the base- and the meta-update step. And we set the Chebyshev approximation order $k$ and the number of meta-framelets $r$ as $k=4$ and $r=4$. We conduct the experiments on varying GNN architectures, with varying parameter budgets (e.g. model size). The experiments are conducted on a single 40G A100 GPU.
>
> As shown in Table 1 and Table 2, given the same parameter budget, our MM-FGCN achieves significantly higher performance than GCN and GAT within a moderate computational overhead. When compared to SVD-GCN and GraphSage, our model achieves higher performance in both training and inference speed and accuracy.

---

> ### Author Response · Authors · 2023-11-20
> **Response to Concern 1. (Part 2/2)**
>
> | Model | \# parameters | Acc. ($\uparrow$) | Time/Epoch ($\downarrow$) | \# parameters | Acc. ($\uparrow$) | Time/Epoch ($\downarrow$) | \# parameters | Acc. ($\uparrow$) | Time/Epoch ($\downarrow$) |
> |-------------------|-----------|---------|-----------------------------|-----------|---------|-----------------------------|----------|----------|-----------------------------|
> | GCN          |   10,246  | 80.4  |**0.012**  | 51,238       | 81.2          | **0.016**               |  100,204  | 82.0               | **0.018**               |
> | GAT           | 10,124   | 79.6             | 0.021       | 50,422                 | 81.7               | 0.022             | 101,684          | 82.5               | 0.024                       |
> | GraphSage [4]    | 10,076    | 78.9               | 0.248                | 51,682        | 80.9               | 0.281             | 107,216          | 81.9               | 0.315                       |
> | SVD-GCN [15]      | 10,578   | 79.6               | 0.236  |           53,412          | 81.4               | 0.254             | 106,452          | 82.5               | 0.261                     |
> | MM-FGCN (ours) w/o meta-training | 10674  |  82.2 | 0.048        | 52,546               | 83.7               | 0.050               | 101,046        | 84.0               | 0.053                       |
> | MM-FGCN (ours)   |  10,674 | **82.6**       | 0.054          | 52,546             | **84.4**       | 0.058         | 101,046              | **84.5**       | 0.61                       |
>
> **Table 1**: Comparison of accuracy and **training** runtime (in seconds) on Cora datasets. Although our model's runtime is slightly larger than GCN and GAT, our method remains faster than GraphSage and SVD-GCN.
>
>
> | Model | \# parameters | Acc. ($\uparrow$) | Inference time ($\downarrow$) | \# parameters | Acc. ($\uparrow$) | Inference time ($\downarrow$) | \# parameters | Acc. ($\uparrow$) | Inference time ($\downarrow$) |
> |-------------------|----------|----------|--------------------------------|------------|--------|--------------------------------|---------|-----------|--------------------------------|
> | GCN          | 10,246     | 80.4               | **0.011**             | 51,238     | 81.2               | **0.015**       | 100,204           | 82.0               | **0.017**                  |
> | GAT        | 10,124       | 79.6               | 0.019              | 50,422            | 81.7               | 0.020              | 101,684            | 82.5               | 0.023                          |
> | GraphSage [4]    | 10,076    | 78.9               | 0.231                 | 51,682         | 80.9               | 0.275                 | 107,216         | 81.9               | 0.302                          |
> | SVD-GCN [15]     | 10,578     | 79.6               | 0.227                |53,412          | 81.4               | 0.247                | 106,452          | 82.5               | 0.258                          |
> | MM-FGCN (ours)  | 10,674  | **82.6**       | 0.047             | 52,546             | **84.4**       | 0.050             | 101,046             | **84.5**       | 0.052                          |
>
> **Table 2**: Comparison of accuracy and **inference** time (in seconds) under different parameter sizes on the Cora dataset.

---

> ### Author Response · Authors · 2023-11-20
> **Response to Concern 2.**
>
> **Concern 2. Additional experiments on OGBN datasets.**
>
> *Weakness 3.* Following the previous point, it seems it's hard to use the model on large graphs. (ogbn tasks)
>
> *Question 2.* Would it be possible to evaluate the model on larger datasets like ogbn? I did observe ogbg-molhiv, but I'm referring to graphs with a larger number of nodes rather than the total number of graphs.
>
> **Response to Concern 2. MM-FGCN achieves SOTA on OGBN dataset.**
>
>
> Our MM-FGCN can be applied to large graphs, e.g. OGBN datasets, and it achieves a high performance.
>
> We conduct experiments on the OGBN-arxiv and OGBN-product datasets, and the results are listed in the following table.
>
> To train our MM-FGCN, for OGBN-arxiv, we use the 128-dimensional feature vector as the node feature of each node; for OGBN-product, we use the 100-dimensional feature vector as each node's feature. It is worth noting that some state-of-the-art methods (such as SimTeG+TAPE+RevGAT, TAPE+RevGAT, and SimTeG+TAPE+GraphSAGE) adopt the raw title and abstract form the OGBN-arxiv and detailed product descriptions from OGBN-product as the node features, and achieve better results due to using a good NLP module in extracting node features. Since we mainly compare the performance of the GNN module, we neglect the gain from using NLP modules such as GIANT-XRT or TAPE, etc. Instead, we directly use the preprocessed feature vectors provided in both datasets on our model and our baseline models.
>
> **Training details**
>
> To train the OGBN-Arxiv, we use the full-batch training. In the OGBN-products dataset, we use neighbor sampling with size = 10 (similar to UniMP [10]) to sample the subgraph during training. Other hyperparameter settings follow our experiments on other datasets as elaborated in the Appendix. We show the corresponding results in the following tables.
>
> | Method     | Test Acc.          | Val. Acc.          |
> |------------|--------------------|--------------------|
> | MLP        | 55.50 $\pm$ 0.23   | 57.65 $\pm$ 0.12   |
> | NODE2VEC [1]   | 70.07 $\pm$ 0.13   | 71.29 $\pm$ 0.13   |
> | GRAPHZOOM [2]  | 71.18 $\pm$ 0.18   | 72.20 $\pm$ 0.07   |
> | P&L + C&S [3]  | 71.26 $\pm$ 0.01   | 73.00 $\pm$ 0.01   |
> | GRAPHSAGE [4]  | 71.49 $\pm$ 0.27   | 72.77 $\pm$ 0.17   |
> | GCN [5]        | 71.74 $\pm$ 0.29   | 73.00 $\pm$ 0.17   |
> | DEEPERGCN [6]  | 71.92 $\pm$ 0.17   | 72.62 $\pm$ 0.14   |
> | SIGN [7]       | 71.95 $\pm$ 0.11   | 73.23 $\pm$ 0.06   |
> | GAAN [8]       | 71.97 $\pm$ 0.18   | –                  |
> | UFGCONV-R [9]  | 71.97 $\pm$ 0.12   | 73.21 $\pm$ 0.05   |
> | UniMP [10]      | 73.11 $\pm$ 0.21| 74.50 $\pm$ 0.05|
> | Exphormer [11]  | 72.44 $\pm$ 0.28   |  –                  |
> | DRGAT [12]      | 74.16 $\pm$ 0.07| 75.34 $\pm$ 0.02|
> | Ours       | **74.25 $\pm$ 0.15**| **75.46 $\pm$ 0.08**|
>
> **Table 3**: Experiment on the OGBN-Arxiv dataset.
>
>
> | Method               | Test Acc.           | Val. Acc.           |
> |----------------------|---------------------|---------------------|
> | MLP                  | 55.50 $\pm$ 0.23    | 57.65 $\pm$ 0.12    |
> | GCN-Cluster          | 78.97 $\pm$ 0.36    | 92.12 $\pm$ 0.09    |
> | GAT-Cluster          | 79.23 $\pm$ 0.78    | 89.85 $\pm$ 0.22    |
> | GAT-NeighborSampling | 79.45 $\pm$ 0.59    | -                   |
> | GraphSAINT [13]          | 80.27 $\pm$ 0.26    | -                   |
> | DeeperGCN  [6]          | 80.90 $\pm$ 0.20    | 92.38 $\pm$ 0.09    |
> | UniMP [10]               | 82.56 $\pm$ 0.31    | 93.08 $\pm$ 0.17    |
> | AGDN  [14]               | 83.34 $\pm$ 0.27    | 92.29 $\pm$ 0.10    |
> | Ours                 | **84.03 $\pm$ 0.23**| **93.57 $\pm$ 0.12**|
>
> **Table 4: Experiment on the OGBN-Products dataset.**
>
>
> The experiment results show that our method achieves the highest performance among various state-of-the-art baselines on both datasets.
>
> For the OGBN-Arxiv dataset in which we use full-batch for training, training one epoch takes 2.86 seconds with meta-learning on average, and the inference takes 2.65 seconds. Such a high speed demonstrates the capability of our model in training on large graph datasets.

---

> ### Author Response · Authors · 2023-11-20
> **Response to Concern 3.**
>
> **Concern 3. A detailed introduction to meta-learning is needed.**
>
> **Response to Concern 3. Meta-learning in MM-FGCN**
>
> The following discussions will be added to the Appendix.
>
> **How meta-learning works?**
>
> Following [16], in this paper, Meta-learning is understood as ‘learning-to-learn’, which refers to the process of improving the base learning algorithm over multiple learning episodes, i.e. the base- and meta-learning episodes. During base-learning, a base learning algorithm solves the base task, such as graph / node classification. During meta-learning, a meta algorithm updates the base learning algorithm such that the model learns to improve an meta objective.
>
> In this paper, the base-objective refers to ‘learning a graph / node classifier using features based on a framelet transform parameterized by $\omega$’, as shown in Equation (8) and Equation (9). We aim to design a meta-objective that refers to ‘finding a good $\omega$ such that the $\omega$-base-objective can be achieved well’. To this end, as shown in the second term of Equation (8), we set the meta-objective to be ‘finding the $\omega$ with the lowest classification loss, for any given network backbone parameter $\theta$’. The intuition behind this is, for any model backbone, we always want to choose the most compatible $\omega$-framelet transform, such that the model backbone $\theta$ can fully release its power, achieving a classification loss. By combining the base- and meta-objectives, we establish the meta-learning training paradigm of MM-FGCN as a bi-level optimization problem in  Equation (8).
>
> Unlike meta-learning models, where the base-learning algorithm is continuously improved during the meta-learning episode, a standard machine learning model is trained on a specific task using fixed, handcrafted base-learning algorithms. For instance, standard GCNs are trained to predict graph labels by minimizing the classification loss, using human-designed features like the graph Fourier transform. On the contrary, our MM-FGCN is able to learn suitable meta-framelet transform that is adaptive to varying graph instances and distributions via the meta-learning paradigm.
>
> **Experimental settings of MM-FGCN.**
>
> In practice, our MM-FGCN shares the same train and test dataset splitting scheme as other baseline methods. We first train the MM-FGCN by meta-training (Algorithm 2) on the training data. Then, we fix the trained parameters $\theta, \xi$, and we evaluate the accuracy of the MM-FGCN on the testing dataset. The dataset splitting in line 3 of Algorithm 2 is done randomly within the training dataset. Hence, in our experiments, the MM-FGCN is evaluated under the same data resource as other baselines.
>
> We split the dataset into main- and meta- dataset to avoid overfitting and enhance generalization performance, and this technique is widely used in other mainstream meta-learning algorithms (e.g. MAML [17]). As shown in Table [5] and [6], our ablation study proves that using the meta-learning paradigm leads to higher performance compared to directly updating all the parameters $\theta, \xi$ .
>
> |                     | Cora | Citeseer | PubMed | Cornell | Texas | Wisconsin | Chameleon | Squirrel |
> |---------------------|------|----------|--------|---------|-------|-----------|-----------|----------|
> | MM-FGCN w/o meta-learning|    82.9 ± 0.5 | 72.2 ± 0.7 | 80.1 $\pm$ 0.2 |    78.2 ± 8.5 | 77.9 ± 9.2 |  82.3 $\pm$ 4.8  |62.6 ± 2.6 | 59.2 ± 2.2     |
> | MM-FGCN           |  **84.4 $\pm$ 0.5** | **73.9 $\pm$ 0.6** | **80.7 $\pm$ 0.2** | **88.9 $\pm$ 8.3** | **86.1 $\pm$ 4.5** | **88.5 $\pm$ 4.1** | **73.97 $\pm$ 2.1** | **67.5 $\pm$ 1.2** |
>
> **Table 5: Comparison of with and without meta-learning framework on the node classification tasks.**
>
> |      | PROTEINS (↑) | Mutagenicity (↑) | D&D (↑) | NCI1 (↑) | Ogbg-molhiv (↑) | QM7 (↓) |
> |-----------------------------|--------------|-------------------|---------|----------|-----------------|---------|
> | MM-FGPool w/o meta-learning |   77.86 $\pm$ 2.53 | 82.32 ± 1.42        |    81.06 ± 1.70 | 75.95 ± 0.91                 |   78.57 $\pm$ 0.73       |   41.65 $\pm$  0.91             |
> | MM-FGPool                   | **78.07 $\pm$ 2.36** | **83.91 $\pm$ 1.32** | **81.51 $\pm$ 1.55** | **78.57 $\pm$ 0.82** | **79.12 $\pm$ 0.85** | **41.19 $\pm$ 0.88** |
>
> **Table 6: Comparison of with and without meta-learning framework on the graph classification tasks.**
>
> **Meta-learning is NOT MAML or Few-shot Learning.**
>
> In contrast to the common narrow definition of “meta-learning” as the MAML (Model Agnostic Meta-Learning) algorithm [17] found in much of the literature, in this paper, "meta-learning" specifically denotes the learning-to-learn paradigm, rather than being restricted to MAML. It's important to note that the MAML algorithm is merely a specific meta-learning technique that is widely used for few-shot learning in classification tasks.

---

> ### Author Response · Authors · 2023-11-20
> **Response to Concern 4. (Part 1/2)**
>
> **Concern 4. The performance is not close to SOTA.**
>
> **Question 4.** The performance is not close to STOA, recent studies show better performance, e.g. Dir-GCN [18] https://arxiv.org/abs/2305.10498 and FSGNN [19] https://arxiv.org/abs/2105.07634.
>
> **Response to Concern 4.**
>
>
> We conduct comprehensive experiments to compare our method with Dir-GCN [18] and FSGNN [19] under the same experimental settings.
>
> **Comparison with FSGNN [19]**
>
> FSGNN adopts a feature selection network to learn the importance value of the feature at each GCN layer, and use that value to scale the magnitude of the features at the corresponding layer. It is worth noting that FSGNN was tested on over 1080 combinations of the hyperparameters, and it provides a set of specific hyper-parameters (e.g. learning rate and weight decay) for each layer of the model. By adopting the same hyper-parameter tuning trick, our model can even achieve higher performance, as shown in the following Table 7.
>
>
>
> |                            | Cora | Citeseer | PubMed | Cornell         | Texas           | Wisconsin        | Chameleon        | Squirrel         |
> |----------------------------|------|----------|--------|-----------------|-----------------|------------------|------------------|------------------|
> | FSGNN [19] (3-hop)          | 80.4 $\pm$ 0.8 | 71.8 $\pm$ 0.5    | 78.9 $\pm$ 0.4 | 87.0 $\pm$ 5.7 | 87.3 $\pm$ 5.5 | 88.4 $\pm$ 3.2 | 78.1 $\pm$ 1.2 | 73.4 $\pm$ 2.1 |
> | FSGNN (8-hop)              | 80.2 $\pm$ 0.9 | 71.9  $\pm$ 0.6   | 78.5 $\pm$ 0.3  | 87.8 $\pm$ 6.2 | 87.3 $\pm$ 5.2 | 87.8 $\pm$ 3.3 | 78.2 $\pm$ 1.2 | 74.1 $\pm$ 1.8 |
> | Ours                       | **84.4 $\pm$ 0.5** | **73.9 $\pm$ 0.6** | **80.7 $\pm$ 0.2** | **88.9 $\pm$ 8.3** | **88.0 $\pm$ 4.9** | **88.9 $\pm$ 3.8** | **78.6 $\pm$ 1.7** | **74.5 $\pm$ 1.4** |
>
> *Table 7: We adopt the hyper-parameter tuning tricks in FSGNN and re-run our experiments on Cornell, Texas, Wisconsin, Chameleon, and Squirrel. For Cora, Citeseer and PubMed, we run the code of FSGNN according to our train/validation/test split: 20 nodes per class for training, 1,000 nodes for testing, and 500 for validation, as elaborated in Section 5.1.*

---

> ### Author Response · Authors · 2023-11-20
> **Response to Concern 4. (Part 2/2)**
>
> **Comparison with Dir-GNN [18]**
>
>
> It is worth noting that the assortative graph datasets: Citeseer-FULL and Cora-ML used in Dir-GNN are completely different from the datasets Cora and Citeseer used in our paper, and the results are not comparable. We first run Dir-GNN under our settings on assortative graph datasets, and the results are shown in the following Table 8.
>
>
>
>
> |             | Cora           | Citeseer       | PubMed         |
> |-------------|----------------|----------------|----------------|
> | Dir-GCN [18] | 80.4   $\pm$ 0.7  | 71.8     $\pm$ 1.0       | 78.9   $\pm$ 0.4       |
> | Dir-GAT [18] | 81.0   $\pm$ 0.9  | 71.1     $\pm$ 0.8       | 79.1  $\pm$  0.3       |
> | Dir-Sage [18]| 80.1    $\pm$ 0.8 | 70.5   $\pm$  1.0        | 77.6   $\pm$ 0.4       |
> | MM-FGCN (Ours)        | **84.4 $\pm$ 0.5** | **73.9 $\pm$ 0.6** | **80.7 $\pm$ 0.2** |
>
> *Table 8: Comparison between Dir-GCN, Dir-GAT, and Dir-Sage with our method on assortative graph datasets: Cora, Citeseer, and PubMed. We conduct the experiments according to our train/validation/test split: 20 nodes per class for training, 1,000 nodes for testing, and 500 for validation, as elaborated in Section 5.1.*
>
>
>
>
> Dir-GNN [18] can be used to extend any Message Passing Neural Network (MPNN) such as GCN, GAT, and GraphSage, and the extended networks are called Dir-GCN, Dir-GAT, and Dir-Sage. The key factor that enhances the performance of Dir-GNN on several disassortative graph neural networks (e.g. Chameleon and Squirrel) is the separate aggregations of the incoming and outgoing edges.
>
> To handle directed graphs, we replace the graph Laplacian of MM-FGCN with the incidence matrix based symmetric Laplacian matrix. To fairly compare our MM-FGCN method with Dir-GCN, Dir-GAT, and Dir-Sage, we implement the Dir-MM-FGCN based on the model used in Dir-GNN by replacing the GNN layers with our MM-FGCN layer. Then, we train our model with Algorithm 2.
>
>
> As shown in Table 9, our model achieves either better or comparable results among all the Dir- baselines, showcasing the advantage of using meta-framelet transform.
>
>
> |             | Cornell         | Texas           | Wisconsin        | Chameleon        | Squirrel         |
> |-------------|-----------------|-----------------|------------------|------------------|------------------|
> | Dir-GCN [18]    | 87.5 $\pm$ 8.7 | 86.9  $\pm$ 5.2 | 86.4 $\pm$ 4.4 |  78.77 $\pm$ 1.7 | 74.4 $\pm$ 0.7 |
> | Dir-GAT [18]    | 76.4 $\pm$ 9.2 | 78.7  $\pm$ 5.1 | 77.2 $\pm$ 4.2 |  71.4 $\pm$ 1.6 | 67.5 $\pm$ 1.1 |
> | Dir-GSage [18]  |       70.5 $\pm$ 9.8  |   73.1  $\pm$ 6.3             |        74.6   $\pm$ 5.8        |      64.5 $\pm$ 2.3 | 46.0 $\pm$ 1.2      |
> | Dir-MM-FGCN (Ours) | **89.7 $\pm$ 8.3**  | **89.2 $\pm$ 4.9**  | **89.8 $\pm$ 3.8**  | **78.9 $\pm$ 1.7**  | **74.0 $\pm$ 0.6**  |
>
> *Table 9: Comparison between Dir-GCN, Dir-GAT, and Dir-Sage with our method Dir-MM-FGCN, which combines the add-on extension of Dir-GNN with our model on disassortative datasets: Cornell, Texas, Wisconsin, Chameleon, and Squirrel.*

---

> ### Author Response · Authors · 2023-11-20
> **References**
>
> [1] Grover, Aditya, and Jure Leskovec. "node2vec: Scalable feature learning for networks." Proceedings of the 22nd ACM SIGKDD international conference on Knowledge discovery and data mining. 2016.
>
> [2] Deng, Chenhui, et al. "Graphzoom: A multi-level spectral approach for accurate and scalable graph embedding." arXiv preprint arXiv:1910.02370 (2019).
>
> [3] Huang, Qian, et al. "Combining label propagation and simple models out-performs graph neural networks." ICLR 2021.
>
> [4] Hamilton, Will, Zhitao Ying, and Jure Leskovec. "Inductive representation learning on large graphs." Advances in neural information processing systems 30 (2017).
>
> [5] Kipf, Thomas N., and Max Welling. "Semi-supervised classification with graph convolutional networks." arXiv preprint arXiv:1609.02907 (2016).
>
> [6] Li, Guohao, et al. "Deepergcn: All you need to train deeper gcns." arXiv preprint arXiv:2006.07739 (2020).
>
> [7] Rossi, Emanuele, et al. "Sign: Scalable inception graph neural networks." arXiv preprint arXiv:2004.11198 7 (2020)
>
> [8] Zhang, Jiani, et al. "Gaan: Gated attention networks for learning on large and spatiotemporal graphs." arXiv preprint arXiv:1803.07294 (2018).
>
> [9] Zheng, Xuebin, et al. "How framelets enhance graph neural networks." arXiv preprint arXiv:2102.06986 (2021).
>
> [10] Shi, Yunsheng, et al. "Masked label prediction: Unified message passing model for semi-supervised classification." arXiv preprint arXiv:2009.03509 (2020).
>
> [11]  Shirzad, Hamed, et al. "Exphormer: Sparse transformers for graphs." arXiv preprint arXiv:2303.06147 (2023).
>
> [12] Zhang, Lei, et al. "DRGCN: Dynamic Evolving Initial Residual for Deep Graph Convolutional Networks." arXiv preprint arXiv:2302.05083 (2023).
>
> [13] Zeng, Hanqing, et al. "Graphsaint: Graph sampling based inductive learning method." arXiv preprint arXiv:1907.04931 (2019).
>
> [14] Sun, Chuxiong, et al. "Adaptive Graph Diffusion Networks." arXiv preprint arXiv:2012.15024 (2020).
>
> [15] Entezari, Negin, et al. "All you need is low (rank) defending against adversarial attacks on graphs." Proceedings of the 13th International Conference on Web Search and Data Mining. 2020.
>
> [16] T. Hospedales, A. Antoniou, P. Micaelli, and A. Storkey. Meta-learning in neural networks: A survey. IEEE Transactions on Pattern Analysis amp; Machine Intelligence, 44(09):5149–5169, sep 2022. ISSN 1939-3539. doi: 10.1109/TPAMI.2021.3079209.
>
> [17] Chelsea Finn, Pieter Abbeel, and Sergey Levine. Model-agnostic meta-learning for fast adaptation of deep networks. In International conference on machine learning, pp. 1126–1135. PMLR, 2017.
>
> [18] Emanuele Rossi et el. Dir-GNN: Graph Neural Networks for Directed Graphs  https://arxiv.org/abs/2305.10498
>
> [19] SK Maurya, X Liu, T Murata. Improving Graph Neural Networks with Simple Architecture Design  https://arxiv.org/abs/2105.07634.

---

> > ### Comment · Reviewer_nDyN · 2023-11-22
> > **response**
> >
> > Thank you for your response. My concerns have been adequately addressed, I increase my score to 6.

---

### Meta-Review · Area_Chair_FLDx · 2023-12-15

**Metareview:**

The paper introduces a new approach to graph representation learning, enabling adaptive multiresolution analysis across a range of graphs. It attains state-of-the-art performance in diverse graph learning tasks. Reviewers acknowledge the significance of this research problem, affirming the novelty and effectiveness of the proposed method in addressing various graph learning tasks. The authors have effectively addressed concerns raised during the rebuttal phase. Therefore, I recommend accepting the paper.

**Justification For Why Not Higher Score:**

I reached this decision by evaluating the contributions and novelty of the work, taking into consideration both the reviews and the responses from the authors.

**Justification For Why Not Lower Score:**

I reached this decision by evaluating the contributions and novelty of the work, taking into consideration both the reviews and the responses from the authors.

---

### Decision · Program_Chairs · 2024-01-16

Accept (poster)